# A General Framework for User-Guided Bayesian Optimization

**Carl Hvarfner**
Lund University
`carl.hvarfner@cs.lth.se`

**Frank Hutter**
University of Freiburg
`fh@cs.uni-freiburg.de`

**Luigi Nardi**
DBtune, Lund University, Stanford University
`luigi.nardi@cs.lth.se`

## Abstract

The optimization of expensive-to-evaluate black-box functions is prevalent in various scientific disciplines. Bayesian optimization is an automatic, general and sample-efficient method to solve these problems with minimal knowledge of the underlying function dynamics. However, the ability of Bayesian optimization to incorporate prior knowledge or beliefs about the function at hand in order to accelerate the optimization is limited, which reduces its appeal for knowledgeable practitioners with tight budgets. To allow domain experts to customize the optimization routine, we propose `ColaBO`, the first Bayesian-principled framework for incorporating prior beliefs beyond the typical kernel structure, such as the likely location of the optimizer or the optimal value. The generality of `ColaBO` makes it applicable across different Monte Carlo acquisition functions and types of user beliefs. We empirically demonstrate `ColaBO`'s ability to substantially accelerate optimization when the prior information is accurate, and to retain approximately default performance when it is misleading.

## 1 Introduction

*Bayesian Optimization* (BO) (Mockus et al., 1978; Jones et al., 1998; Snoek et al., 2012) is a well-established methodology for the optimization of expensive-to-evaluate black-box functions. Known for its sample efficiency, BO has been successfully applied to a variety of domains where laborious system tuning is prominent, such as hyperparameter optimization (Snoek et al., 2012; Bergstra et al., 2011b; Lindauer et al., 2022), neural architecture search (Ru et al., 2021; White et al., 2021), robotics (Calandra et al., 2014; Mayr et al., 2022), hardware design (Nardi et al., 2019; Ejjeh et al., 2022), and chemistry (Griffiths & Hernández-Lobato, 2020).

Typically employing a Gaussian Process (Rasmussen & Williams, 2006) (GP) surrogate model, BO allows the user to specify a prior over functions $p(f)$ via the Gaussian Process kernel, as well as an optional prior over its hyperparameters. Within the framework of the prior, the user can specify expected smoothness, output range and possible noise level of the function at hand, with the hopes of accelerating the optimization if accurate. However, the prior beliefs that can be specified within the framework of the kernel hyperparameters do not span the full range of beliefs that practitioners may possess. For example, practitioners may know which *parts of the input space* tend to work best (Oh et al., 2018; Perrone et al., 2019; Smith, 2018; Wang et al., 2019), know a range or upper bound on the optimal output (Jeong & Kim, 2021; Nguyen & Osborne, 2020) such as a maximal achievable accuracy of 100%, or other properties of the objective, such as preference relations between configurations (Huang et al., 2022). The limited ability of practitioners to interact and collaborate with the BO machinery (Kumar et al., 2022) thus runs the risk of failing to use valuable domain expertise, or alienating knowledgeable practitioners altogether. While knowledge injection beyond what is natively supported by the GP kernel is crucial to further increase the efficiency of

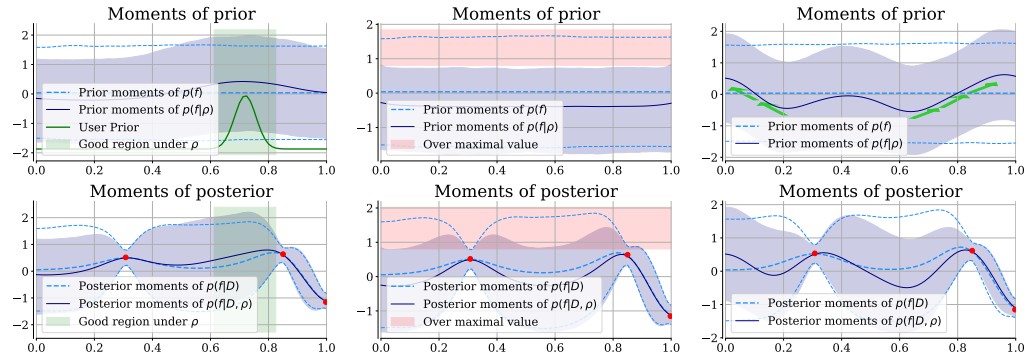

**Figure 1:** Three different `ColaBO` priors: (left) Prior over the optimum $\boldsymbol{x}_*$, and the induced changed in the GP for an optimum located in the green region. (middle) Prior over optimal value, $f^* < 0.8$. (right) Prior over preference relations $f(\boldsymbol{x})_1 \geqslant f(\boldsymbol{x}_2)$ for five preferences (green arrows, e.g. $f(0.0) \geqslant f(0.1) \geqslant f(0.2)$).

Bayesian optimization, so far no current approach allows for the integration of arbitrary types of user knowledge. To address this gap, we propose an intuitive framework that effectively allows the user to reshape the Gaussian process at will to mimic their held beliefs.

Figure 1 illustrates how, for the three aforementioned priors, the GP is reshaped to *faithfully represent* the belief held by the user - whether it be a prior over the optimum (left), optimal value (middle), or preference relations between points (right). Our novel framework for *Collaborative Bayesian Optimization* (`ColaBO`) diverges from the typical assumption of Gaussian posteriors, and is applicable to any Monte Carlo acquisition function (Wilson et al., 2017; 2018; Balandat et al., 2020). Formally, we make the following contributions:

1. We introduce `ColaBO`, a framework for incorporating user knowledge over the optimizer, optimal value and preference relations into Bayesian optimization in the form of an additional prior on the surrogate, orthogonal to the conventional prior on the kernel hyperparameters,

2. We demonstrate that the proposed framework is generally applicable to Monte Carlo acquisition functions, inheriting MC acquisiiton function utility,

3. We empirically show that `ColaBO` accelerates optimization when injected with for priors over optimal location and optimal value.

## 2 BACKGROUND

We outline Bayesian optimization, Gaussian Processes and Monte Carlo (MC) acquisition functions, as well as the concept of a prior over the optimum.

### 2.1 BAYESIAN OPTIMIZATION

We consider the problem of optimizing a black-box function $f$ across a set of feasible inputs $\mathcal{X} \subset \mathbb{R}^d$:

$$\boldsymbol{x}^* \in \arg\max_{\boldsymbol{x} \in \mathcal{X}} f(\boldsymbol{x}). \tag{1}$$

We assume that $f(\boldsymbol{x})$ is expensive to evaluate and can potentially only be observed through a noise-corrupted estimate, $y_{\boldsymbol{x}}$, where $y_{\boldsymbol{x}} = f(\boldsymbol{x}) + \epsilon, \epsilon \sim \mathcal{N}(0, \sigma_\epsilon^2)$ for some noise level $\sigma_\epsilon^2$. In this setting, we wish to maximize $f$ in an efficient manner. Bayesian optimization (BO) aims to globally maximize $f$ by an initial design and thereafter sequentially choosing new points $\boldsymbol{x}_n$ for some iteration $n$, creating the data $\mathcal{D}_n = \mathcal{D}_{n-1} \cup \{(\boldsymbol{x}_n, y_n)\}$ (Brochu et al., 2010; Shahriari et al., 2016; Garnett, 2022). After each new observation, BO constructs a probabilistic surrogate model $p(f|\mathcal{D}_n)$ (Snoek et al., 2012; Hutter et al., 2011; Bergstra et al., 2011a; Müller et al., 2023) and uses that surrogate to build an acquisition function $\alpha(\boldsymbol{x}; \mathcal{D}_n)$ which selects the next query.

## 2.2 GAUSSIAN PROCESSES

When constructing the surrogate, the most common choice is a *Gaussian process* (GP) (Rasmussen & Williams, 2006). The GP utilizes a covariance function $k$, which encodes a prior belief for the smoothness of $f$, and determines how previous observations influence prediction. Given observations $\mathcal{D}_n$ at iteration $n$, the Gaussian posterior $p(f|\mathcal{D}_n)$ over the objective is characterized by the posterior mean $\mu_n(\boldsymbol{x}, \boldsymbol{x}')$ and (co-)variance $\Sigma_n(\boldsymbol{x}, \boldsymbol{x}')$ of the GP:

$$\mu_n(\boldsymbol{x}) = \mathbf{k}_n(\boldsymbol{x})^\top (\mathbf{K}_n + \sigma_\epsilon^2 \mathbf{I})^{-1} \mathbf{y}, \quad \Sigma_n(\boldsymbol{x}, \boldsymbol{x}') = k(\boldsymbol{x}, \boldsymbol{x}') - \mathbf{k}_n(\boldsymbol{x})^\top (\mathbf{K} + \sigma_\epsilon^2 \mathbf{I})^{-1} \mathbf{k}_n(\boldsymbol{x}'),$$

where $(\mathbf{K}_n)_{ij} = k(\boldsymbol{x}_i, \boldsymbol{x}_j)$, $\mathbf{k}_n(\boldsymbol{x}) = [k(\boldsymbol{x}, \boldsymbol{x}_1), \ldots, k(\boldsymbol{x}, \boldsymbol{x}_n)]^\top$ and $\sigma_\epsilon^2$ is the noise variance. For applications in BO and beyond, samples from the posterior are required either directly for optimization (Eriksson et al., 2019) through Thompson sampling (Thompson, 1933), or to estimate auxiliary quantities of interest (Hernández-Lobato et al., 2015; Neiswanger et al., 2021; Hvarfner et al., 2023). For a finite set of $k$ query locations ($\boldsymbol{X} = \boldsymbol{x}_1, \ldots, \boldsymbol{x}_k$), the classical method of generating samples is via a location-scale transform of Gaussian random variables, $f(\boldsymbol{X}) = \mu_n(\boldsymbol{X}) + \boldsymbol{L}\boldsymbol{\epsilon}$, where $\boldsymbol{L}$ is the Cholesky decomposition of $\boldsymbol{K}$ and $\boldsymbol{\epsilon} \sim \mathcal{N}(0, \boldsymbol{I})$. Unfortunately, the classic approach is intrinsically non-scalable, incurring a $\mathcal{O}(k^3)$ cost due to the aforementioned matrix decomposition.

## 2.3 DECOUPLED POSTERIOR SAMPLING

To remedy the issue of scalability in posterior sampling, $\mathcal{O}(k)$ weight-space approximations based on Random Fourier Features (RFF) (Rahimi & Recht, 2007) obtain approximate (continuous) function draws $\tilde{f}(\boldsymbol{x}) = \sum_{i=1}^m w_i \phi_i(\boldsymbol{x})$, where $\phi_i(\boldsymbol{x}) = \frac{2}{\ell}(\boldsymbol{\psi}_i^\top \boldsymbol{x} + b_i)$. The random variables $w_i \sim \mathcal{N}(0, 1)$, $b_i \sim \mathcal{U}(0, 2\pi)$, and $\boldsymbol{\psi}_i$ are sampled proportional to the spectral density of $k$.

While achieving scalability, the seminal RFF approach by Rahimi & Recht (2007) suffers from the issue of variance starvation (Mutny & Krause, 2018; Wang et al., 2018; Wilson et al., 2020). As a remedy, Wilson et al. (2020) decouple the draw of functions from the approximate posterior $p(\tilde{f}|\mathcal{D})$ into a more accurate draw from the prior $p(\tilde{f})$, followed by a deterministic data-dependent update:

$$(\tilde{f}|\mathcal{D})(\boldsymbol{x}) \stackrel{d}{=} \underbrace{\tilde{f}(\boldsymbol{x})}_{\text{draw from prior}} + \underbrace{\mathbf{k}_n(\boldsymbol{x})^\top (\mathbf{K}_n + \sigma_\epsilon^2 \mathbf{I})^{-1}(\mathbf{y} - \tilde{f}(\boldsymbol{x}) - \boldsymbol{\epsilon})}_{\text{deterministic update}} \tag{2}$$

Eq. 2 deviates from the distribution-first approach that is typically prevalent in GPs in favor of a variable-first approach utilizing Matheron's rule (Journel & Huijbregts, 1976).

## 2.4 MONTE CARLO ACQUISITION FUNCTIONS

Acquisition functions act on the surrogate model to quantify the utility of a point in the search space. They encode a trade-off between exploration and exploitation, typically using a greedy heuristic to do so. A simple and computationally cheap heuristic is Expected Improvement (EI) (Jones et al., 1998; Bull, 2011). For a noiseless function and a current best observation $y_n^*$, the EI acquisition function is $\alpha_{EI}(\boldsymbol{x}) = \mathbb{E}_{y_{\boldsymbol{x}}}[(y_n^* - y_{\boldsymbol{x}})^+]$. For noisy problem settings, a noise-adapted variant of EI (Letham et al., 2018) is frequently considered, where both the incumbent $y_n^*$ and the upcoming query $y_{\boldsymbol{x}}$ are substituted for the non-observable noiseless incumbent $f_n^*$ and noiseless upcoming query $f_{\boldsymbol{x}}$. Other frequently used acquisition functions are the Upper Confidence Bound (UCB) (Srinivas et al., 2012), Probability of Improvement (PI) (Kushner, 1964) and Knowledge Gradient (KG) (Frazier et al., 2009). Information-theoretic acquisition functions consider the mutual information to select the next query $\alpha_{MI}(\boldsymbol{x}) = I((\boldsymbol{x}, y_{\boldsymbol{x}}); *|\mathcal{D}_n)$, where $*$ can entail either the optimum $\boldsymbol{x}_*$ as in (Predictive) Entropy Search (ES/PES) (Hennig & Schuler, 2012; Hernández-Lobato et al., 2014), the optimal value $f_*$ as in Max-value Entropy Search (MES) (Wang & Jegelka, 2017; Moss et al., 2021) or the tuple $(\boldsymbol{x}_*, f_*)$ for Joint Entropy Search (JES) (Hvarfner et al., 2022a; Tu et al., 2022).

All the aforementioned acquisition functions compute expectations $\mathbb{E}_{f_{\boldsymbol{x}}}$ (or alternatively $\mathbb{E}_{y_{\boldsymbol{x}}}$) over some utility $u(f_{\boldsymbol{x}})$ of the output (Wilson et al., 2017; 2018), which typically have simple, or even closed-form, solutions for Gaussian posteriors. However, approximating the expectation through Monte Carlo integration has proven useful in the context of batch optimization (Wilson et al., 2018), efficient acquisition function approximation (Balandat et al., 2020), and non-Gaussian posteriors (Astudillo & Frazier, 2021). By sampling over possible outputs $f_{\boldsymbol{x}}$ and utilizing the reparametrization

trick (Kingma & Welling, 2014; Rezende et al., 2014), utilities $u$ can be easily computed across a larger set of applications and be optimized to greater accuracy.

## 2.5 PRIOR OVER THE OPTIMUM

A prior over the optimum (Souza et al., 2021; Hvarfner et al., 2022b; Mallik et al., 2023) is a user-specified belief $\pi : \mathcal{X} \rightarrow \mathbb{R}$ of the subjective likelihood that a given $\boldsymbol{x}$ is optimal. Formally,

$$\pi(\boldsymbol{x}) = \mathbb{P}\left(\boldsymbol{x} = \arg\max_{\boldsymbol{x}'} f(\boldsymbol{x}')\right). \tag{3}$$

This prior is generally considered to be independent of observed data, but rather a result of previous experimentation or anecdotal evidence. Regions that the user expects to contain the optimum will typically have a high value, but this does not exclude the chance of the user belief $\pi(\boldsymbol{x})$ to be inaccurate, or even misleading. Lastly, we require $\pi$ to be strictly positive in all of $\mathcal{X}$, which suggests that any point included in the search space may be optimal.

## 3 METHODOLOGY

We now introduce `ColaBO`, the first Bayesian-principled BO framework that flexibly allows users to *collaborate* with the optimizer by injecting prior knowledge about the objective that substantially exceeds the type of prior knowledge natively supported by GPs. In Sec. 3.1, we introduce and derive a novel prior over function properties, which yields a surrogate model conditioned on the user belief. Thereafter, in Sec. 3.2, we demonstrate how the hierarchical prior integrates with MC acquisition functions. Lastly, in Sec. 3.3, we state practical considerations to assure the performance of `ColaBO`.

## 3.1 PRIOR OVER FUNCTION PROPERTIES

We consider the typical GP prior over functions $p(f) = \mathcal{GP}(\mu, \Sigma)$, where the characteristics of $f$, such as smoothness and output magnitude, are fully defined by the kernel $k$ (and its associated hyperparameters $\boldsymbol{\theta}$, which are omitted for brevity). We seek to inject an additional, user-defined prior belief over $f$ into the GP, such as the prior over the optimum in Sec. 2.5, $\pi(\boldsymbol{x}) = \mathbb{P}(\boldsymbol{x} = \arg\max_{\boldsymbol{x}'} f(\boldsymbol{x}'))$. By postulating that $\pi$ is accurate, we wish to form a belief-weighted prior - a prior over *functions* where the distribution over the optimum is exactly $\pi(\boldsymbol{x})$. We start by considering the user belief $\pi : \mathcal{X} \rightarrow \mathbb{R}$ from Eq. (3), and extend the definition to involve the integration over $f$, similarly to the Thompson sampling definition of Kandasamy et al. (2018). Formally,

$$\pi(\boldsymbol{x}) = \mathbb{P}\left(\boldsymbol{x} = \arg\max_{\boldsymbol{x}'} f(\boldsymbol{x}')\right) = \int_f \pi(\delta_*(\boldsymbol{x}|f)) p(f) df \tag{4}$$

where $\delta_*(\boldsymbol{x}|f) = 1$, if $\boldsymbol{x} = \arg\max_{\boldsymbol{x}' \in \mathcal{X}} f(\boldsymbol{x}')$, and zero otherwise. As such, $\delta_*(\boldsymbol{x}|f)$ maps a function $f_i \sim p(f)$ to its $\arg\max$, and evaluates whether this $\arg\max$ is equal to $\boldsymbol{x}$.

However, a belief over the optimum, or any other property, of a function $f$ is implicitly a belief over the function $f$ itself. As such, a non-uniform $\pi(\boldsymbol{x})$ should reasonably induce a change in the prior $p(f)$ to reflect the non-uniform optimum. To this end, we introduce an augmented user belief over the optimum $\rho_{\boldsymbol{x}}^* \sim \mathcal{P}_{\boldsymbol{x}}^*$, where $\mathcal{P}_{\boldsymbol{x}}^*$ is the prior over possible user beliefs, and draws are random functions $\rho_{\boldsymbol{x}}^* : \mathcal{X} \rightarrow \mathbb{R}^+$ which themselves take a function $f$ as input, and output a positive real number quantifying the likelihood of a sample $f_i$ under $\pi(\boldsymbol{x})$. Formally, we define $\rho_{\boldsymbol{x}}^*$ as

$$\rho_{\boldsymbol{x}}^*(f) = \mathbb{P}\left(\boldsymbol{x} = \arg\max_{\boldsymbol{x}'} f(\boldsymbol{x}')\right) = \frac{1}{Z_{\rho_{\boldsymbol{x}}^*}} \int_{\mathcal{X}} \delta_*(\boldsymbol{x}|f) \pi(\boldsymbol{x}) d\boldsymbol{x} \tag{5}$$

where the intractible normalizing constant $Z_{\rho_{\boldsymbol{x}}^*}$ arises from the fact that the integrated density $\pi(\boldsymbol{x})$ acts on a finite-dimensional *property* of $f$, and not $f$ itself. Under $\rho_{\boldsymbol{x}}^*(f)$, functions whose $\arg\max$ lies in a high-density region under $\pi$ will be assigned a higher probability. Notably, the definition in 5 can extend to other properties of $f$ as well: a user belief $p_{f_*}$ over the optimal value $f_*$ analogously yields a belief over functions $\rho_{f_{\boldsymbol{x}}}^*(f)$:

$$\rho_{f_{\boldsymbol{x}}}^*(f) = \mathbb{P}\left(\boldsymbol{x} = \max_{\boldsymbol{x}'} f(\boldsymbol{x}')\right) = \frac{1}{Z_{\rho_{f_{\boldsymbol{x}}}^*}} \int_{f_{\boldsymbol{x}}} \delta_*(\boldsymbol{x}|f) p_{f*}(y) df_{\boldsymbol{x}}. \tag{6}$$

It is worthwhile to reflect on the meaning of $\rho(f)$, and how beliefs over function properties propagate to $p(f)$. Concretely, if the user belief $\rho_{f_{\boldsymbol{x}}}^*(f)$ asserts that the maximal value lies within $C_1 < \max f < C_2$, the resulting distribution over $f$ should only contain functions whose max falls within this range. Using rejection sampling, functions which disobey this criterion are filtered out, which yields the posterior $p(f|\rho)$. Having defined and exemplified how user beliefs impact the prior over functions $p(f)$, the role of $\rho$ as a likelihood should be apparent: given a prior over functions $p(f)$ and a user belief over functions $\rho(f)$ which places a probability on all possible draws $f_i\ p(f)$, we can form a belief-weighted prior $p(f|\rho)\propto p(f)\rho(f)$. Thus, we introduce the formal definition of a user belief over a function property:

**Definition 3.1** (User Belief over Functions). *The user belief over functions $\rho(f)\propto\frac{p(f|\rho)}{p(f)}$.*

As the subsequent derived methodology applies independently of the specific property of $f$ that a prior is placed on, we will henceforth consider a belief over a general function property $\rho$. Having defined the role of $\rho$ and the posterior over functions it produces, a natural question arises: *How is $p(f|\rho)$ updated once observations $\mathcal{D}$ are obtained?*

Since the data $\mathcal{D}$ is independent of the prior (the data generation process is intrinsically unaffected by the belief held by the user), application of Bayes' rule yields the following posterior $p(f|\mathcal{D}, \rho)$,

$$p(f|\mathcal{D}, \rho) = \frac{p(\mathcal{D}, \rho|f)p(f)}{p(\mathcal{D}, \rho)} = \frac{p(\mathcal{D}|f)p(\rho|f)p(f)}{p(\mathcal{D})p(\rho)} = \frac{p(f|\rho)}{p(f)}p(f|\mathcal{D}) \propto \rho(f)p(f|\mathcal{D}), \qquad (7)$$

where the right side of the proportionality in Eq. 7 suggests an intuitive generation process for samples $(f|\mathcal{D}, \rho)$ to approximate the density $p(f|\mathcal{D}, \rho)$. Utilizing the pathwise update from Eq. 2, we note that given an approximate draw $\tilde{f}$ from the prior, the subsequent data-dependent update is deterministic. Recalling Eq. 2 and assuming independence between $\rho$ and $\mathcal{D}$, $\rho$ only affects the draw from the prior, whereas $\mathcal{D}$ only affects the update. Consequently, we obtain

$$(\tilde{f}|\mathcal{D}, \rho)(\boldsymbol{x}) \overset{d}{=} \underbrace{(\tilde{f}|\rho)(\boldsymbol{x})}_{\text{draw from prior}} + \underbrace{\mathbf{k}_n(\boldsymbol{x})^\top(\mathbf{K}_n + \sigma_\epsilon^2\mathbf{I})^{-1}(\mathbf{y} - (\tilde{f}|\rho)(\boldsymbol{x}) - \boldsymbol{\epsilon})}_{\text{deterministic update}}, \qquad (8)$$

where $(\tilde{f}|\rho) \sim p(f)\rho(\tilde{f})$ are once again obtained using rejection sampling on draws from $p(\tilde{f})$. Figure 2 displays this in detail: given the typical GP prior over functions *and* a user belief over the optimum, we obtain a distribution over functions $p(\tilde{f}|\rho_{\boldsymbol{x}}^*)$ before having observed any data (top right). Samples from the approximate prior $p(\tilde{f})$ (light blue) are re-sampled proportionally to their probability of occurring under the prior $\rho_{\boldsymbol{x}}^*(\tilde{f})$ in green, leaving samples $(\tilde{f}|\rho_{\boldsymbol{x}}^*)$ in navy blue, which are highly probable under $\rho_{\boldsymbol{x}}^*$. Once data is obtained, these samples are updates according to Eq. 8, which preserves the shape of the samples far away from observed data and yields the desired posterior.

### 3.2 PRIOR-WEIGHTED MONTE CARLO ACQUISITION FUNCTIONS

Naturally, neither the belief-weighted prior $p(f|\rho)$ nor the belief-weighted posterior $p(f|\mathcal{D}, \rho)$ have a closed-form expression. Both are inherently non-Gaussian for non-uniform beliefs. As such, we resort to MC acquisition functions to compute utilities that are amenable to BO. In the subsequent section, we focus on the prevalent acquisition functions `EI`, and `MES`.

**Expected Improvement** The computation of the MC-`EI` within the `ColaBO` framework requires only minor adaptations of the original MC acquisition function. By definition, MC-`EI` assigns utility $u$ as $u_{\text{EI}}(f(\boldsymbol{x})) = \max(f_n^* - f(\boldsymbol{x}), 0)$, which yields

$$\alpha_{\text{EI}}(\boldsymbol{x}; \mathcal{D}) = \mathbb{E}_{f_{\boldsymbol{x}}|\mathcal{D}}[u_{\text{EI}}(f_{\boldsymbol{x}})] \approx \qquad (9)$$

$$\sum_\ell \max(f_n^* - f_{\boldsymbol{x}}^{(\ell)}, 0), \ f_{\boldsymbol{x}}^{(\ell)} \sim p(f(\boldsymbol{x})|\mathcal{D}). \qquad (10)$$

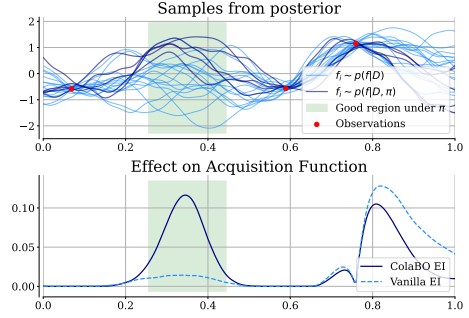

**Figure 3:** (Top) Draws from $p(f|\mathcal{D})$ (light blue) and $p(f|\rho, \mathcal{D})$ with a prior $\rho$ located in the green region. (Bottom) Vanilla MC-`EI` and `ColaBO` MC-`EI`, resulting from computing the acquisition function from sample draws from $p(f|\rho, \mathcal{D})$.

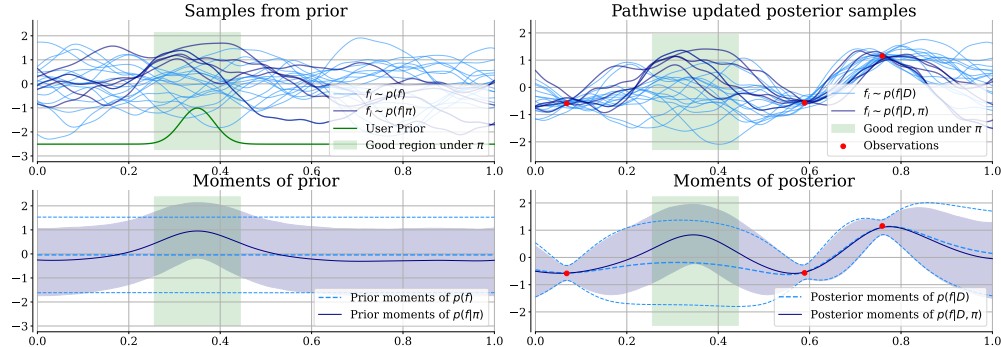

**Figure 2:** (Top left) Draws from the prior $p(f)$ (light blue) and the belief-weighted prior $p(f|\rho)$ whose members are likely to have their optimum within the green region. (Top right) Pathwise updated draws based on observed data. As the green region is distant from the observed data, samples are almost unaffected by the data in this region. (Bottom left) Exact mean and standard deviation $(\mu_{\boldsymbol{x}}, \sigma_{\boldsymbol{x}})$ of $p(f)$ and estimated mean and standard deviation of $p(f|\rho)$. (Bottom right) Exact $p(f|\mathcal{D})$ and estimated $p(f|\rho, \mathcal{D})$. As $p(f|\rho)$ constitutes of functions whose optimum is located within the green region the resulting model has a higher mean and lower variance within this region. Moreover, $p(f|\rho)$ globally displays lower upside variance compared to the vanilla GP.

Utilizing rejection sampling, we can compute the MC-`EI` under the `ColaBO` posterior accordingly,

$$\alpha_{\texttt{EI}}(\boldsymbol{x}; \mathcal{D}, \rho) = \mathbb{E}_{f_{\boldsymbol{x}}|\mathcal{D},\rho}[u_{\texttt{EI}}(f_{\boldsymbol{x}})] \propto \tag{11}$$

$$\int_f u_{\texttt{EI}}(f_{\boldsymbol{x}})\rho(f)p(f|\mathcal{D})df \approx \sum_\ell \rho(f^{(\ell)})\max(f_n^* - f_{\boldsymbol{x}}^{(\ell)}, 0), \quad f_{\boldsymbol{x}}^{(\ell)} \sim p(f(\boldsymbol{x})|\mathcal{D}), \tag{12}$$

wherein samples in Eq. 12 are drawn from the prior, retained with probability $\rho(f^{(\ell)})/\max \rho$, and pathwise updated. In Figure 3, we demonstrate how `ColaBO-EI` differs from MC-`EI` for an identical posterior as in Figure 2. By computing $\alpha_{\texttt{EI}}$ from samples biased by $\rho$, `ColaBO` substantially directs the search towards good regions under $\rho$. Derivations for `PI` and `KG` are analogous to that of `EI`.

**Max-Value Entropy Search**  We derive a `ColaBO-MES` acquisition function by first considering the definition of the entropy, $\mathrm{H}[p(y_{\boldsymbol{x}}|\mathcal{D})] = \mathbb{E}_{y_{\boldsymbol{x}}|\mathcal{D}}[-\log p(y_{\boldsymbol{x}}|\mathcal{D})]$. When considering the belief-weighted posterior, we further condition the posterior on $\rho$ and obtain

$$\alpha_{\texttt{MES}}(\boldsymbol{x}) = \mathbb{E}_{f_*|\mathcal{D},\rho}\left[\mathbb{E}_{y_{\boldsymbol{x}}|\mathcal{D},\rho,f_*}[\log p(y_{\boldsymbol{x}}|\mathcal{D}, \rho, f_*)]\right] - \mathbb{E}_{y_{\boldsymbol{x}}|\mathcal{D},\rho}[\log p(y_{\boldsymbol{x}}|\mathcal{D}, \rho)] \tag{13}$$

$$\propto \mathbb{E}_{f_*|\mathcal{D},\rho}\left[\mathbb{E}_{f_{\boldsymbol{x}}|\mathcal{D},\rho}[\mathbb{E}_{y_{\boldsymbol{x}}|f_{\boldsymbol{x}}}[\log p(y_{\boldsymbol{x}}|f_{\boldsymbol{x}}, \rho, f_*)]]\right] - \mathbb{E}_{f_{\boldsymbol{x}}|\mathcal{D},\rho}[\mathbb{E}_{y_{\boldsymbol{x}}|f_{\boldsymbol{x}}}[\log p(y_{\boldsymbol{x}}|f_{\boldsymbol{x}}, \rho)]] \tag{14}$$

$$\approx \frac{1}{Z_J}\sum_{j=1}^J \sum_{\ell=1}^L \sum_{k=1}^K \log p(y_{\boldsymbol{x}}^{(k)}|f_{\boldsymbol{x}}^{(\ell)}, f_*^{(j)})\rho(f^{(\ell)})\rho(f^{(j)}) - \sum_{\ell=1}^L \sum_{k=1}^K \log p(y_{\boldsymbol{x}}^{(k)}|f_{\boldsymbol{x}}^{(\ell)})\rho(f^{(\ell)}), \tag{15}$$

where $Z_J$ is a normalizing constant $\sum_J \rho(f^{(j)})$ brought on by sampling optimal values, $y_{\boldsymbol{x}}|f_{\boldsymbol{x}}$ can trivially be obtained by sampling Gaussian noise $\varepsilon \sim \mathcal{N}(0, \sigma_\varepsilon^2)$ to a noiseless observation $f_{\boldsymbol{x}}|\mathcal{D}$ in the innermost expectation, and $f_{\boldsymbol{x}}$ and $f_*$ are obtained through the pathwise sampling procedure outlined in Eq. 8. The samples are evaluated on $p((y_{\boldsymbol{x}}|f_{\boldsymbol{x}}), (y_{\boldsymbol{x}}|f_{\boldsymbol{x}}, f_*))$. As evident by Eq. 15, $\rho$ affects the posterior distribution of both the observations $y_{\boldsymbol{x}}$ and the optimal values $f_*$. `PES` and `JES` are derived analogously. However, these acquisition function require conditioning on additional, simulated data and consequently, additional pathwise updates, to compute.

### 3.3 PRACTICAL CONSIDERATIONS

`ColaBO` introduces additional flexibility to MC-based BO acquisition functions. The `ColaBO` framework deviates from vanilla (q-)MC acquisition functions (Wilson et al., 2017; Balandat et al., 2020) by utilizing approximate sample functions from the posterior, as opposed to pointwise draws from the posterior predictive and the reparametrization trick (Kingma & Welling, 2014; Rezende et al., 2014). `ColaBO` holds three shortcomings not prevalent in vanilla MC acquisition functions: (1) it cannot utilize Quasi-MC in the draws from the predictive posterior (only in the RFF weights),

---

**Algorithm 1** `ColaBO` iteration

---

1: **Input:** User prior $\rho$, number of function samples $L$, current data $\mathcal{D}$
2: **Output:** Next query location $\boldsymbol{x}'$.
3: **for** $\ell \in \{1, \ldots, L\}$ **do**
4: $\quad \rho^{(\ell)} = \rho(\tilde{f}^{(\ell)}; n), \tilde{f}^{(\ell)} \sim p(\tilde{f})$      ▷ Sample functions and evaluate on $\pi$
5: $\quad (\tilde{f}^{(\ell)}|\mathcal{D}) = \texttt{PathwiseUpdate}(\tilde{f}^{(\ell)}, \mathcal{D})$     ▷ Per-sample update as in Eq. 8
6: **end for**
7: $p(\tilde{f}|\mathcal{D}, \rho) \approx \sum_{\ell} \rho^{(\ell)}(\tilde{f}^{(\ell)}|\mathcal{D})$      ▷ Form MC estimate of posterior
8: $\boldsymbol{x}' = \arg\max_{\boldsymbol{x} \in \mathcal{X}} \mathbb{E}_{p(\tilde{f}|\mathcal{D},\rho)}[u(\tilde{f}_{\boldsymbol{x}})]$      ▷ Maximize MC acquisition

---

(2) it cannot fix the base samples (Balandat et al., 2020) drawn from the posterior for acquisition function consistency across the search space, and (3) the RFF approximation of $p(f)$ introduces bias. This approximation error is substantially more pronounced for the Matérn 5/2-kernel than the squared exponential, leaving ColaBO best suited for the latter. In Sec. 4.1, we empirically display the impact of these shortcomings. While acquisition function optimization no longer enjoys improved accuracy resulting from reparametrization, the acquisition function can still benefit from the fact that `ColaBO` backpropagates through quantities computed as sums of smooth functions.

## 4 RESULTS

We evaluate the performance of `ColaBO` on various tasks, using priors over the optimum $\pi_{\boldsymbol{x}_*}$ obtained from known optima on synthetic tasks, as well as from prior work (Mallik et al., 2023) on realistic tasks. We consider two variants of `ColaBO`: one using `LogEI` (Ament et al., 2023), a numerically stable, smoothed `logsumexp` transformation of `EI` with analogous derivation, and one variant using `MES`. We benchmark against the vanilla variants of each acquisition function, as well as $\pi$BO (Hvarfner et al., 2022b) and decoupled Thompson sampling Thompson (1933); Wilson et al. (2020). All acquisition functions are implemented in BoTorch (Balandat et al., 2020) using a squared exponential kernel and MAP hyperparameter estimation. We present experiments with a Matérn-5/2 (Matérn, 1960) kernel in App. C.1. Unless stated otherwise, all methods are initialized with the mode of the prior followed by 2 Sobol samples. The experimental setup is outlined in Appendix B, and our code is publicly available at `https://github.com/hvarfner/colabo`.

### 4.1 APPROXIMATION QUALITY OF THE COLABO FRAMEWORK

Firstly, we demonstrate the approximation quality of `ColaBO` *without* user priors to assert its accuracy compared to a vanilla MC acquisition function. To facilitate comparison, we randomly sample 10 points on the Hartmann (3D) function, and optimize `LogEI` with a large budget. We subsequently optimize `ColaBO-LogEI` on the same set of points and compare the $\arg\max$ to the solution found by the gold standard. Figure 4 displays the (log10) Euclidian distance between the $\arg\max$ of `LogEI` and its `ColaBO` variant. We note that, for small amounts ($\leqslant 256$) of posterior samples, the error induced by RFF bias is relatively low, which is evidenced by all RFF variants being roughly equal in distance to the true acquisition function optimizer.

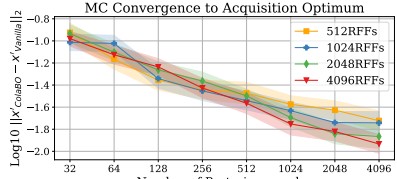

**Figure 4:** Mean and $1/4$ standard deviation of MC-induced errors of `ColaBO-LogEI` relative vanilla `LogEI` as measured by the distance to the $\arg\max$ of the acquisition function on Hartmann (3D) on 10 randomly sampled points for 40 seeds.

### 4.2 SYNTHETIC FUNCTIONS WITH KNOWN PRIORS

We adapt a similar evaluation protocol to Hvarfner et al. (2022b), and evaluate `ColaBO` for two types of user beliefs for synthetic tasks: well-located and poorly located priors over the optimal location, designed to emulate a well-informed and poorly-informed practitioner, respectively. The well-located prior is offset by a small (10%) amount from the optimum, and the poorly located prior is maximally offset, while retaining its mode inside the search space. Complete details on the priors can be found

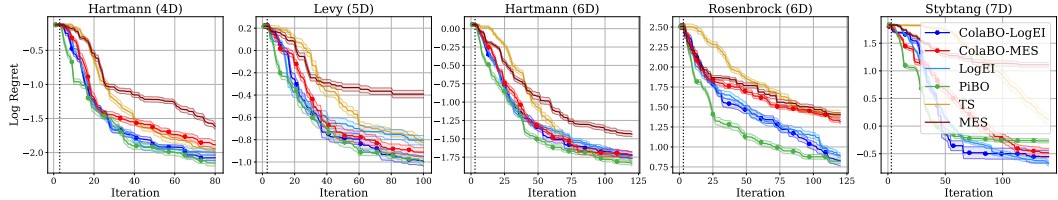

**Figure 5:** Performance on synthetic functions with well-located priors. Both `ColaBO-LogEI` and `ColaBO-MES` offer drastic speed-ups over their vanilla variants, and offer similar performance to $\pi$BO. The ranking of `ColaBO` acquisition functions are generally consistent with their respective vanilla variants. This is most prominent on Rosenbrock (6D), where `ColaBO-MES` struggles similarly to vanilla `MES`.

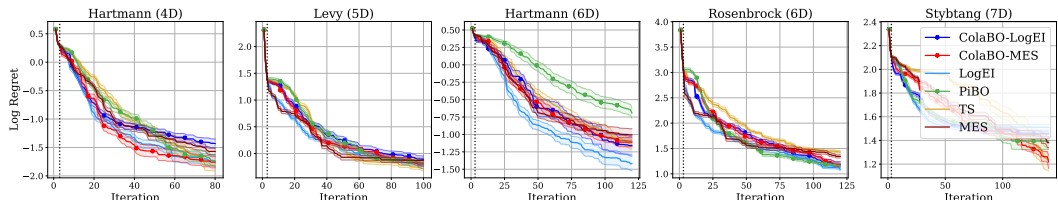

**Figure 6:** Performance on poorly located priors. `ColaBO` acquisition functions are more robust than $\pi$BO, as it frequently recovers the performance of the vanilla acquisition function before the total budget is depleted. `ColaBO-LogEI` struggles marginally on Hartmann (6D). `ColaBO-MES` recovers the baseline on all tasks.

in Appendix B.3. On well-located priors, both `ColaBO-LogEI` and `ColaBO-MES` demonstrate substantially improved performance relative to their vanilla counterparts, comparable to $\pi$BO on all benchmarks. On poorly located priors, `ColaBO` demonstrates superior robustness, recovering the performance of the vanilla acquisition function within the maximal budget of $20D$ iterations and clearly outperforming $\pi$BO, which more frequently misled by the poor prior. In Appendix C.2, we also demonstrate `ColaBO` utilizing (accurate) beliefs over the optimal value: similarly to Figure 5, `ColaBO` yields increased efficiency relative to baselines, albeit not as substantial. Moreover, we demonstrate its usage with batch evaluations on well-located priors in Sec. C.3, showing that the drop in performance from batching evaluations is marginal at worst.

## 4.3 Hyperparameter Tuning tasks

Lastly, we evaluate `ColaBO` on three $4D$ deep learning HPO tasks from the PD1 (Wang et al., 2023) benchmarking suite. While the optima for these tasks are ultimately unknown, we utilize the priors provided in MF-Prior-Bench [1] (Mallik et al., 2023), which are intended to provide a good starting point for further optimization. To emulate a realistic HPO setting, we consider a smaller optimization budget of $10D$ iterations, and initalize all methods that utilize user beliefs with only one initial sample, that being the mode of the prior. The two `ColaBO` variants perform best in this evaluation, producing the best terminal performance on two tasks (CIFAR, LM1B), with all methods being tied on the third (CIFAR). `ColaBO` demonstrates consistent speed-ups compared to its vanilla counterparts, surpassing the terminal performance of the baseline within a third of the budget on CIFAR and LM1B. In App. A, we benchmark on 5 tasks from LBBench (Zimmer et al., 2020), displaying similar performance.

## 5 Related Work

In BO, auxiliary prior information can be conveyed in multiple ways. We outline meta learning/transfer learning for BO based on data from previous experiments, and data-less approaches.

**Learning from Previous Experiments** Transfer learning and meta learning for BO aims to automatically extract and use knowledge from prior executions of BO by pre-training the model on

---

[1] https://github.com/automl/mf-prior-bench

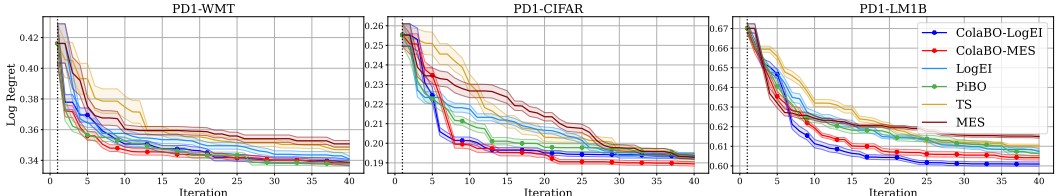

**Figure 7:** Performance on the 4D PD1 hyperparameter tuning tasks of various deep learning pipelines. `ColaBO` drastically accelerates optimization initially, finding configurations with close to terminal performance quickly. $\pi$BO offers competitive performance, but lacks the rapid initial progress of `ColaBO` on CIFAR and LM1B.

data acquired from previous executions (Swersky et al., 2013; Wistuba et al., 2015; Perrone et al., 2019; Feurer et al., 2015; 2018; Rothfuss et al., 2021a;b; Wistuba & Grabocka, 2021; Feurer et al., 2022). Typically, meta- and transfer learning exploit relevant previous data for training the GP for the current task while retaining predictive uncertainty to account for imperfect task correlation.

**Expert Priors over Function Optimum** Few previous works have proposed to inject explicit prior distributions over the location of an optimum into BO. In these cases, users explicitly define a prior that encodes their beliefs on where the optimum is more likely to be located. Bergstra et al. (2011a) suggest an approach that supports prior beliefs from a fixed set of distributions, which affects the very initial stage of optimization. However, this approach cannot be combined with standard acquisition functions. BOPrO (Souza et al., 2021) employs a similar structure that combines the user-provided prior distribution with a data-driven model into a pseudo-posterior. From the pseudo-posterior, configurations are selected using the EI acquisition function, using the formulation in Bergstra et al. (2011a). $\pi$BO (Hvarfner et al., 2022b) suggests a general-purpose prior-weighted acquisition function, where the influence of the prior decreases over time. They provide convergence guarantees for when the framework is applied to the EI acquisition function. While effective, none of these approaches act on the surrogate model in a Bayesian-principled fashion, but strictly as heuristics. Moreover, they solely focus on priors over optimal inputs, thus offering less utility than `ColaBO`.

**Priors over Optimal Value** Similarly few works have addressed the issue of auxilliary knowledge of the optimal value. Both Jeong & Kim (2021) and Nguyen & Osborne (2020) propose altering the GP and accompanying it with tailored acquisition functions. Jeong & Kim (2021) employ variational inference, proposing distinct variational families depending on the type of knowledge pertaining to the optimal value. Nguyen & Osborne (2020) use a parabolic transformation of the output space to ensure an upper bound is preserved. Unlike `ColaBO`, neither of these methods is general enough to accompany arbitrary user priors to guide the optimization.

## 6 Conclusion, Limitations and Future Work

We presented `ColaBO`, a flexible BO framework that allows practitioners to inject beliefs over function properties in a Bayesian-principled manner, allowing for increased efficiency in the BO procedure. `ColaBO` works across a collection of MC acquisition functions, inheriting their flexibility in batch optimization and ability to work with non-Gaussian posteriors. It demonstrates competitive performance for well-located priors, using them to substantially accelerate optimization. Moreover, it retains approximately baseline performance when applied to detrimental priors, demonstrating greater robustness than $\pi$BO. `ColaBO` crucially relies on multiple steps of MC. While flexible, this approach drives computational expense in order to assert sufficient accuracy, requiring tens of seconds per evaluation to achieve desired accuracy, depending on the size of the benchmark. Moreover, obtaining draws from $\rho_x^*$ scales exponentially in the dimensionality of the prior. While practitioners are unlikely to specify priors over more than a handful of variables, `ColaBO` may become impractical when priors of higher dimensionality are employed. Paths for future work could involve more accurate and efficient sampling procedures (Lin et al., 2023) from the belief-weighted prior, as well as variational (Titsias, 2009) or pre-trained Müller et al. (2022); Müller et al. (2023) approaches to obtain a representative belief-biased model with an analytical posterior. This would likely bring down the runtime of `ColaBO` and broaden its potential use. Lastly, applying `ColaBO` to multi-fidelity optimization (Kandasamy et al., 2016; Mallik et al., 2023) offers an additional avenue for increased efficiency which would further increase its viability on costly deep learning pipelines.

## ACKNOWLEDGEMENTS

We thank the anonymous reviewers for their valuable contributions. Luigi Nardi was supported in part by affiliate members and other supporters of the Stanford DAWN project — Ant Financial, Facebook, Google, Intel, Microsoft, NEC, SAP, Teradata, and VMware. Carl Hvarfner, Erik Hellsten and Luigi Nardi were partially supported by the Wallenberg AI, Autonomous Systems and Software Program (WASP) funded by the Knut and Alice Wallenberg Foundation. Luigi Nardi was partially supported by the Wallenberg Launch Pad (WALP) grant Dnr 2021.0348. Frank Hutter acknowledges support through TAILOR, a project funded by the EU Horizon 2020 research and innovation programme under GA No 952215, by the Deutsche Forschungsgemeinschaft (DFG, German Research Foundation) under grant number 417962828, by the state of Baden-Württemberg through bwHPC and the German Research Foundation (DFG) through grant no INST 39/963-1 FUGG, and by the European Research Council (ERC) Consolidator Grant "Deep Learning 2.0" (grant no. 101045765). The computations were also enabled by resources provided by the Swedish National Infrastructure for Computing (SNIC) at LUNARC partially funded by the Swedish Research Council through grant agreement no. 2018-05973. Funded by the European Union. Views and opinions expressed are however those of the author(s) only and do not necessarily reflect those of the European Union or the ERC. Neither the European Union nor the ERC can be held responsible for them.

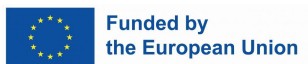

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

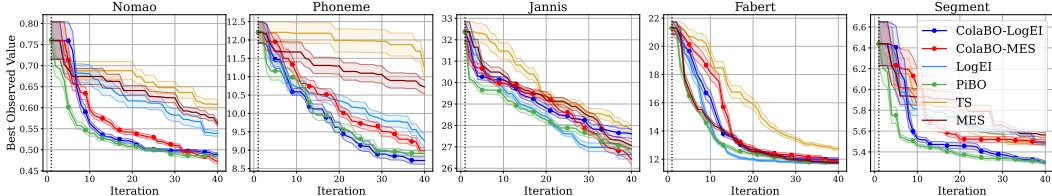

**Figure 8:** Performance on the 6D LCBench hyperparameter tuning tasks of various deep learning pipelines. ColaBO substantially improves on the non-prior baselines for 3 out of five tasks. $\pi$BO performs best on aggregate, and achieves the best acceleration in performance at early iterations.

## A  LCBench Benchmarking

We evaluate all methods on five deep learning tasks (6D) from the LCBench (Zimmer et al., 2020) suite, utilizing priors from MF-Prior-Bench. The chosen tasks were the five tasks with available priors of the best (good) strength, as per the benchmark suite. Figure 8 shows the performance of all methods on the LCBench tasks. ColaBO improves substantially on the baseline approaches for 3 out of 5 tasks. $\pi$BO is the overall best-performing method, followed by ColaBO-LogEI.

## B  Experimental Setup

### B.1  Model

We outline the model used and the budget allocated to the various MC approximations involved with ColaBO. For all experiments, we utilize MAP estimation of the hyperparameters, and update the hyperparameters at every iteration of BO. All hyperparameters - lengthscale, outputscale and observation noise ($\boldsymbol{\theta} = \{\ell, \sigma_\varepsilon^2, \sigma_f^2\}$) are given conventional $\mathcal{LN}(0,1)$ prior, applied on normalized inputs and standardized outputs. Furthermore, we fit the constant $c$ of the mean function, assigning it a $\mathcal{N}(0,1)$ prior as well. In Tab. 1, we display the parameters of the MC approximations for various tasks. *No. $f$* is the maximal number of functions used in the MC computation of the acquisition function. *No. Reamples* is the number of initial posterior draws maximally used for the re-sampling of functions from the posterior $p(f|\rho)$. Lastly, . *No. $f_*$* is the number of optimal values used in the computation of ColaBO-MES.

| Task | No. $f$ | No. RFFs | No. Resamples | No. $f_*$ |
|---|---|---|---|---|
| Synthetic Good | 768 | 2048 | $1.5 * 10^5$ | 32 |
| Synthetic Bad | 768 | 2048 | $1.5 * 10^5$ | 32 |
| PD1 | 512 | 4096 | $2 * 10^5$ | 32 |
| Appendix | 512 | 1024 | $10^5$ | 32 |

**Table 1:** Budget-related parameters of the Monte Carlo approximations for all tasks.

### B.2  Benchmarks

We outline the benchmarks used, their search spaces and the amount of synthetic noise added. When adding noise, we intend for the ratio of noise variance to total output range to be approximately equal across benchmarks.

### B.3  Priors

For synthetic benchmarks, the approximate optima of all included functions can be obtained in advance. Thus, the correctness of the prior is ultimately known in advance. For a function of dimensionality $d$ with optimum at $\boldsymbol{x}_*$, the well-located prior is constructed by sampling an offset

| Task | Dimensionality | $\sigma_\epsilon$ | Search space |
|---|---|---|---|
| Hartmann (4D) | 4 | 0.25 | $[0,1]^D$ |
| Levy (5D) | 5 | 0.5 | $[-5,5]^D$ |
| Hartmann (6D) | 6 | 0.25 | $[0,1]^D$ |
| Rosenbrock (6D) | 6 | 5 | $[-2.048, 2.048]^D$ |
| Stybtang (7D) | 7 | 1 | $[-4,4]^D$ |

**Table 2:** Benchmarks used for the Bayesian optimization experiments.

direction $\epsilon$ and scaling the offset by a dimensionality- and quality-specific term $c(d,q) = q\sqrt{d}$. For the well-located prior on synthetic tasks, we use $q = 0.1$, which implies that the optimum is located 10% of the distance across the search space away from the optimum, and construct a Gaussian prior as

$$\pi_{\boldsymbol{x}_*}(\boldsymbol{x}) \sim \mathcal{N}(\boldsymbol{x}_* + c_d \boldsymbol{\epsilon}/||\boldsymbol{\epsilon}||, \sigma_s), \quad \boldsymbol{\epsilon} \sim \mathcal{N}(0, \boldsymbol{I}). \tag{16}$$

with $\sigma_s = 25\%$ for all tasks and prior qualities. For our 20 runs of the well-located prior, this procedure yields us 20 unique priors per quality type, with identical offsets from the true optimum. No priors with a mode outside the search space were allowed, such priors were simply replaced. For the misinformed priors, we set $q = 1$, guaranteeing that the mode of the prior will be outside of the search space, and subsequently relocating to the edge of the search space by its shortest path. Priors for all tasks are displayed in Tab. 3. For the PD1 tasks, the location for the priors were obtained from MF-Prior-Bench( https://github.com/automl/mf-prior-bench). However, these priors require offsetting in order to not be too strong, thus making subsequent BO obsolete. PD1 priors are provided in $[0,1]$-normalized space for simplicity.

| Task | Location | Offset, good | Offset, bad | $\sigma_s$ |
|---|---|---|---|---|
| Hartmann (4D) | $[0.19, 0.19, 0.56, 0.26]$ | $0.1\sqrt{D}$ | max | 0.25 |
| Levy (5D) | $[1]^D$ | $1\sqrt{D}$ | max | 2.5 |
| Hartmann (6D) | $[0.20, 0.15, 0.48, 0.28, 0.31, 0.66]$ | $0.1\sqrt{D}$ | max | 0.25 |
| Rosenbrock (6D) | $[1]^D$ | $0.4096\sqrt{D}$ | max | 1.024 |
| Stybtang (7D) | $[-2.9]^D$ | $0.8\sqrt{D}$ | max | 2 |
| PD1-WMT | $[0.90, 0.69, 0.02, 0.97]$ | $0.05\sqrt{D}$ | N/A | 0.25 |
| PD1-CIFAR | $[1, 0.80, 0.0, 0.0]$ | $0.05\sqrt{D}$ | N/A | 0.25 |
| PD1-LM1B | $[0.91, 0.67, 0.36, 0.85]$ | $0.05\sqrt{D}$ | N/A | 0.25 |

**Table 3:** $\pi_{\boldsymbol{x}_*}$ for synthetic BO tasks of both prior qualities and PD1.

## C  ADDITIONAL EXPERIMENTS

We provide complementary experiments to those introduced in the main paper. Firstly, we display results when `ColaBO` is used with a prior $\pi_{f_*}$ over the optimal value in Sec. C.2. In Sec. C.3, we demonstrate `ColaBO`:s extensibility to batch evaluations, seamlessly extending the work of (Wilson et al., 2017).

### C.1  SYNTHETIC MATERN KERNEL EXPERIMENTS

We evaluate `ColaBO` and all baselines on the synthetic tasks with a Matern-5/2 kernel and the good user belief over the optimum. We note that roughly half of all $\pi$BO runs struggle with numerical instability from iteration 60 onwards, which produces stagnation in performance and infrequent gains.

### C.2  MAX-VALUE PRIORS

We evaluate `ColaBO` with priors over the optimal value $\pi_{f_*}$ in Figure 10. For each task, we place a Gaussian prior over the optimal value, centering it exactly at the optimal value. Notably, such a prior

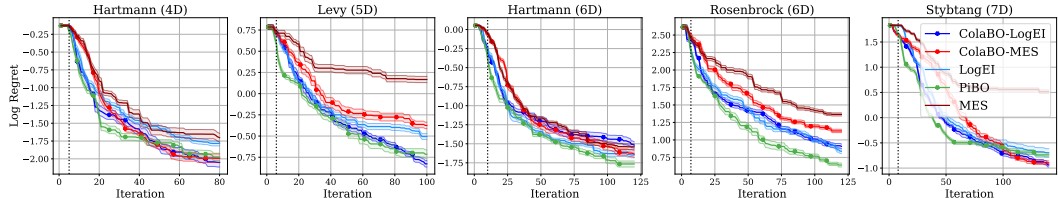

**Figure 9:** `ColaBO` on the synthetic tasks with a Matern kernel. Due to the difficulty of the RFF approximation, `ColaBO-LogEI` struggles on Hartmann (6D), and `ColaBO` performance is marginally worse on aggregate.

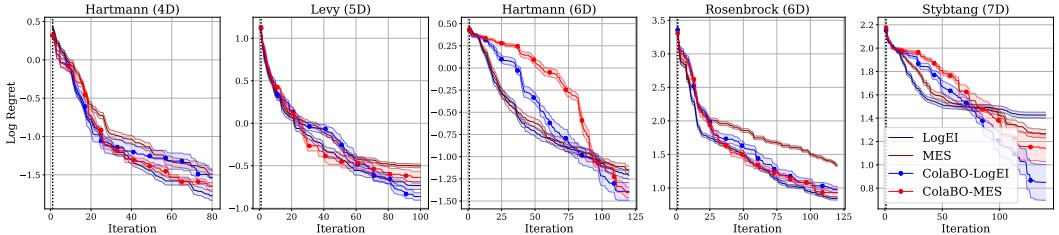

**Figure 10:** `ColaBO` with priors over the optimal value. Terminal performance substantially increases on 3 out of 5 benchmarks (Levy, Hartmann (6D), Stybtang), and is approximately preserved on the final two. `ColaBO-MES` improves marginally more than `ColaBO-LogEI` when utilizing a prior $\pi_{f*}$ over the optimal value.

substantially influences the exploration-exploitation trade-off; if the prior suggests that the incumbent has a value close to the optimal one, we are encouraged to exploit as samples with well-above-optimal values in exploratory will be discarded. Conversely, we are heavily encouraged to explore if the current best observation holds a value that we believe is far from optimal. On Hartmann (6D), we can see this behavior at play. Initial performance is poorer for `ColaBO` than their respective baselines, presumably due to above-average exploration, but terminal performance is better.

### C.3  BATCH EVALUATIONS

We evaluate `ColaBO` on batch evaluations, utilizing the sequential greedy technique for MC acquisition functions from Wilson et al. (2018). Drop-off from sequential to batch evaluations is not evident from the plots, as ordering between sequential and batch varies with the benchmark. While unpredictable, we speculate that the altered exploration-exploitation trade-off provided by the batched acquisition function is occasionally beneficial in the presence of auxilliary user beliefs $\pi_{x*}$.

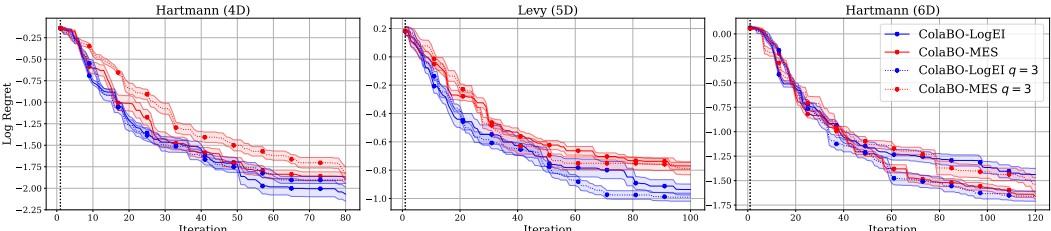

**Figure 11:** $q = 1$ (sequential) and $q = 3$ batch evaluation on a subset of synthetic functions with well-located priors for `ColaBO-LogEI` and `ColaBO-MES`. Total function evaluations are plotted for both sequential and batched variants, leaving them with the same number of total function evaluations.

