# A General Framework for User-Guided Bayesian Optimization

## Abstract

The optimization of expensive-to-evaluate black-box functions is prevalent in various scientific disciplines. Bayesian optimization is an automatic, general and sample-efficient method to solve these problems with minimal knowledge of the the underlying function dynamics. However, the ability of Bayesian optimization to incorporate prior knowledge or beliefs about the function at hand in order to accelerate the optimization is limited, which reduces its appeal for knowledgeable practitioners with tight budgets. To allow domain experts to customize the optimization routine, we propose `ColaBO`, the first Bayesian-principled framework for incorporating prior beliefs beyond the typical kernel structure, such as the likely location of the optimizer or the optimal value. The generality of `ColaBO` makes it applicable across different Monte Carlo acquisition functions and types of user beliefs. We empirically demonstrate `ColaBO`'s ability to substantially accelerate optimization when the prior information is accurate, and to retain approximately default performance when it is misleading.

## 1 Introduction

*Bayesian Optimization* (BO) is a well-established methodology for the optimization of expensive-to-evaluate black-box functions. Known for its sample efficiency, BO has been successfully applied to a variety of domains where laborious system tuning is prominent, such as hyperparameter optimization (Snoek et al., 2012; Bergstra et al., 2011b; Lindauer et al., 2022), neural architecture search (Ru et al., 2021; White et al., 2021), robotics (Calandra et al., 2014; Mayr et al., 2022), hardware design (Nardi et al., 2019; Ejjeh et al., 2022), and chemistry (Griffiths & Hernández-Lobato, 2020).

Typically employing a Gaussian Process (Rasmussen & Williams, 2006) (GP) surrogate model, BO allows the user to specify a prior over functions $p(f)$ via the Gaussian Process kernel, as well as an optional prior over its hyperparameters. Within the framework of the prior, the user can specify expected smoothness, output range and possible noise level of the function at hand, with the hopes of accelerating the optimization if accurate. However, the prior beliefs that can be specified within the framework of the kernel hyperparameters do not span the full range of beliefs that practitioners may possess. For example, practitioners may know which *parts of the input space* tend to work best (Oh et al., 2018; Perrone et al., 2019; Smith, 2018; Wang et al., 2019), know a range or upper bound on the optimal output (Jeong & Kim, 2021; Nguyen & Osborne, 2020) such as a maximal achievable accuracy of 100% , or other properties of the objective, such as preference relations between configurations (Huang et al., 2022). The limited ability of practitioners to interact and collaborate with the BO machinery (Kumar et al., 2022) thus runs the risk of failing to use valuable domain expertise, or alienating knowledgeable practitioners altogether.

While knowledge injection beyond what is natively supported by the GP kernel is crucial to further increase the efficiency of Bayesian optimization, so far no approach exists that allows the integration of knowledge regarding arbitrary properties of the function at hand. To address this gap, we propose an intuitive framework that is agnostic to the user's prior, and that effectively allows the user to reshape the Gaussian process at will to mimic their held beliefs. Our novel framework for *Collaborative Bayesian Optimization* (`ColaBO`) diverges from the typical assumption of Gaussian posteriors, and

is applicable to any Monte Carlo acquisition function (Wilson et al., 2017; 2018; Balandat et al., 2020). Formally, we make the following contributions:

1. We introduce `ColaBO`, a framework for incorporating arbitrary user knowledge into Bayesian optimization in the form of an additional prior on the surrogate, orthogonal to the conventional prior on the kernel hyperparameters,

2. We demonstrate that the proposed framework is generally applicable to Monte Carlo acquisition functions and various priors over arbitrary properties of the objective function,

3. We empirically show that `ColaBO` substantially accelerates optimization when the prior is accurate, while being only marginally hindered by a severely misleading prior.

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

)$. We observe that $\pi$ is an (unnormalized) density for function draws $f$, since $f$ can be mapped to its $\arg\max$. Subsequently, it can serve as an input to $\pi(\boldsymbol{x})$. In turn, this observation reveals the posterior $p(f|\pi)$ through which $\pi(\boldsymbol{x})$ is defined. Formally,

$$\pi(\boldsymbol{x}) = \mathbb{P}\left(\boldsymbol{x} = \arg\max_{\boldsymbol{x}'} f(\boldsymbol{x}')\right) = \int_f \delta_*(\boldsymbol{x}|f) p(f|\pi) df \qquad (4)$$

where the Dirac delta operator $\delta_*(\boldsymbol{x}|f)$ maps $f$ to its $\arg\max$.

It is worthwhile to reflect on the meaning of $p(f|\pi)$ in greater detail. For a user-specified belief $\pi(f)$ over a *property* of $f$, such as the location of its $\arg\max \pi_{\boldsymbol{x}_*}(f)$ (Hvarfner et al., 2022b), the value of its optimum $\max \pi_{f_*}(f)$ (Nguyen & Osborne, 2020), or pairwise preference relations $\boldsymbol{x}' \succcurlyeq \boldsymbol{x}$, $\pi_{\succcurlyeq \boldsymbol{x}}(f)$ (Huang et al., 2022), the belief over such properties propagate to the belief over $f$ itself. Concretely, if the user belief $\pi_{f_*}(f)$ asserts that the maximal value lies within $C_1 < \max f^* < C_2$, the resulting distribution over $f$ should only contain functions whose $\max$ falls within this range. Using rejection sampling, functions which disobey this criterion are filtered out, which yields the posterior $p(f|\pi_{f_*})$.

Having exemplified how user beliefs impact the prior over functions $p(f)$, we hereby introduce the formal definition of a user belief over a function property:

**Definition 3.1** (User Belief over Functions)**.** *The user belief over functions is* $\pi(f) \propto p(f|\pi)/p(f)$.

Intuitively, this definition states that the unnormalized density $\pi(f)$ specifies how likely a function $f$ is to come from $p(f|\pi)$, relative to the vanilla $p(f)$. Moreover, it allows us to view $\pi$ through the lens of a likelihood. Having defined the role of $\pi$ and the posterior over functions it produces, a natural question arises: *How is the belief-weighted posterior $p(f|\pi)$ updated once observations $\mathcal{D}$ are obtained?*

Since the data $\mathcal{D}$ is independent of the prior (the data generation process is intrinsically unaffected by the belief held by the user), successive application of Bayes' rule yields the following posterior conditioned on $\mathcal{D}$ and $\pi$,

$$p(f|\mathcal{D}, \pi) = \frac{p(\mathcal{D}, \pi|f)p(f)}{p(\mathcal{D}, \pi)} = \frac{p(\mathcal{D}|f)p(\pi|f)p(f)}{p(\mathcal{D})p(\pi)} = \frac{p(f|\pi)}{p(f)} p(f|\mathcal{D}) \propto \pi(f)p(f|\mathcal{D}), \qquad (5)$$

where the right side of the proportionality in Eq. 5 suggests an intuitive generation process for samples $(f|\mathcal{D}, \pi)$ to approximate the density $p(f|\mathcal{D}, \pi)$. Utilizing the pathwise update from Eq. 2, we note that given an approximate draw $\tilde{f}$ from the prior, the subsequent data-dependent update is deterministic. Recalling Eq. 2 and assuming independence between $\pi$ and $\mathcal{D}$, $\pi$ only affects the draw from the prior, whereas $\mathcal{D}$ only affects the update. Consequently, we obtain

$$(\tilde{f}|\mathcal{D}, \pi)(\boldsymbol{x}) \overset{d}{=} \underbrace{(\tilde{f}|\pi)(\boldsymbol{x})}_{\text{draw from prior}} + \underbrace{\mathbf{k}_n(\boldsymbol{x})^\top (\mathbf{K}_n + \sigma_\epsilon^2 \mathbf{I})^{-1}(\mathbf{y} - (\tilde{f}|\pi)(\boldsymbol{x}) - \boldsymbol{\epsilon})}_{\text{deterministic update}}, \qquad (6)$$

where $(\tilde{f}|\pi) \sim p(f)\pi(\tilde{f})$ are once again obtained using rejection sampling on draws from $p(\tilde{f})$. Figure 1 displays this in detail: given the typical GP prior over functions *and* a user belief over the optimum, we obtain a posterior over functions $p(\tilde{f}|\pi)$ before having observed any data (top right). Samples from the approximate prior $p(\tilde{f})$ (light blue) are re-sampled proportionally to their

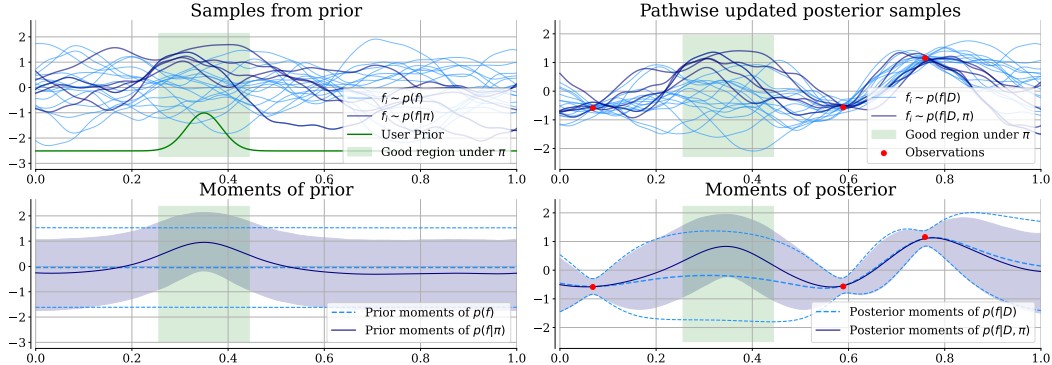

**Figure 1:** (Top left) Draws from the prior $p(f)$ (light blue) and the belief-weighted prior $p(f|\pi)$ whose members are likely to have their optimum within the green region. (Top right) Pathwise updated draws based on observed data. As the green region is distant from the observed data, samples are almost unaffected by the data in this region. (Bottom left) Exact mean and standard deviation ($\mu_x, \sigma_x$) of $p(f)$ and estimated mean and standard deviation of $p(f|\pi)$. (Bottom right) Exact $p(f|\mathcal{D})$ and estimated $p(f|\pi, \mathcal{D})$. As $p(f|\pi)$ constitutes of functions whose optimum is located within the green region the resulting model has a higher mean and lower variance within this region. Moreover, $p(f|\pi)$ globally displays lower upside variance compared to the vanilla GP.

probability of occurring under the prior $\pi_{x_*}(\tilde{f})$ in green, leaving samples $(\tilde{f}|\pi_{x_*})$ in navy blue, which are highly probable under $\pi_{x_*}$. Once data is obtained, these samples are updates according to Eq. 6, which preserves the shape of the samples far away from observed data and yields the desired posterior.

### 3.2 Prior-weighted Monte Carlo Acquisition Functions

Naturally, neither the belief-weighted prior $p(f|\pi)$ nor the belief-weighted posterior $p(f|\mathcal{D}, \pi)$ have a closed-form expression. Both are inherently non-Gaussian for non-uniform beliefs. As such, we resort to MC acquisition functions to compute utilities that are amenable to BO. In the subsequent section, we focus on the prevalent acquisition functions `EI`, and `MES`.

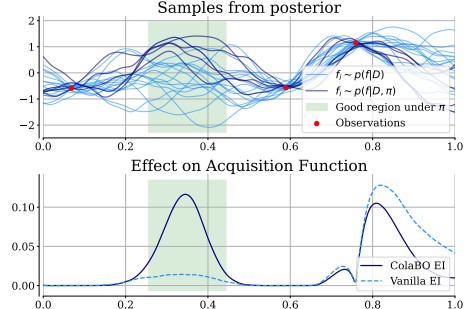

**Expected Improvement** The computation of the Monte Carlo approximation of `EI` within the `ColaBO` framework requires only minor adaptations of the original MC acquisition function. By definition, MC-`EI` assigns utility $u$ as $u_{\text{EI}}(f(x)) = \max(f_n^* - f(x), 0)$, which yields

$$\alpha_{\text{EI}}(x; \mathcal{D}) = \mathbb{E}_{f_x|\mathcal{D}}[u_{\text{EI}}(f_x)] \approx \qquad (7)$$

$$\sum_\ell \max(f_n^* - f_x^{(\ell)}, 0), \quad f_x^{(\ell)} \sim p(f(x)|\mathcal{D}). \qquad (8)$$

**Figure 2:** (Top) Draws from the posterior $p(f|\mathcal{D})$ (light blue) and the belief-weighted posterior $p(f|\pi, \mathcal{D})$ with a prior $\pi$ located in the green region. (Bottom) Vanilla MC-`EI` and `ColaBO` MC-`EI`, resulting from computing the acquisition function from many sample draws from $p(f|\pi, \mathcal{D})$.

Utilizing importance sampling, we can compute the MC-`EI` under the `ColaBO` posterior accordingly,

$$\alpha_{\text{EI}}(x; \mathcal{D}, \pi) = \mathbb{E}_{f_x|\mathcal{D}, \pi}[u_{\text{EI}}(f_x)] = \qquad (9)$$

$$\int_f u_{\text{EI}}(f_x)\pi(f)p(f|\mathcal{D})df \approx \sum_\ell \pi(f^{(\ell)})\max(f_n^* - f_x^{(\ell)}, 0), \quad f_x^{(\ell)} \sim p(f(x)|\mathcal{D}), \qquad (10)$$

wherein samples in Eq. 10 are drawn from the prior, evaluated on $\pi$, and pathwise updated. In Figure 2, we demonstrate how `ColaBO-EI` differs from vanilla MC-`EI` for an identical posterior as in Figure 1. By computing $\alpha_{\text{EI}}$ from samples biased by $\pi_{x_*}$, `ColaBO` substantially biases the search towards regions that are good under $\pi_{x_*}$. Derivations for `PI` and `KG` are analogous to that of `EI`.

---

**Algorithm 1** `ColaBO` iteration

---

1: **Input:** User prior $\pi$, number of function samples $L$, current data $\mathcal{D}$, tempering factor $\beta$
2: **Output:** Next query location $\boldsymbol{x}'$.
3: **for** $\ell \in \{1, \dots, L\}$ **do**
4:      $\pi^{(\ell)} = \pi(\tilde{f}^{(\ell)}; \beta, n), \tilde{f}^{(\ell)} \sim p(\tilde{f})$      $\triangleright$ `Sample functions and evaluate on` $\pi$
5:      $(\tilde{f}^{(\ell)}|\mathcal{D}) = \texttt{PathwiseUpdate}(\tilde{f}^{(\ell)}, \mathcal{D})$      $\triangleright$ `Per-sample update as in Eq. 6`
6: **end for**
7: $p(\tilde{f}|\mathcal{D}, \pi) \approx \sum_{\ell} \pi^{(\ell)}(\tilde{f}^{(\ell)}|\mathcal{D})$      $\triangleright$ `Form MC estimate of posterior`
8: $\boldsymbol{x}' = \arg\max_{\boldsymbol{x} \in \mathcal{X}} \mathbb{E}_{p(\tilde{f}|\mathcal{D}, \pi)}[u(\tilde{f}_{\boldsymbol{x}})]$      $\triangleright$ `Maximize MC acquisition`

---

**Max-Value Entropy Search** We derive a `ColaBO-MES` acquisition function by first considering the definition of the entropy, $\mathrm{H}[p(y_{\boldsymbol{x}}|\mathcal{D})] = \mathbb{E}_{y_{\boldsymbol{x}}|\mathcal{D}}[-\log p(y_{\boldsymbol{x}}|\mathcal{D})]$. When considering the belief-weighted posterior, we further condition the posterior on $\pi$ and obtain

$$\alpha_{\mathrm{MES}}(\boldsymbol{x}) = \mathbb{E}_{f_*|\mathcal{D}, \pi} \left[ \mathbb{E}_{y_{\boldsymbol{x}}|\mathcal{D}, \pi, f_*}[\log p(y_{\boldsymbol{x}}|\mathcal{D}, \pi, f_*)] \right] - \mathbb{E}_{y_{\boldsymbol{x}}|\mathcal{D}, \pi}[\log p(y_{\boldsymbol{x}}|\mathcal{D}, \pi)] \tag{11}$$

$$\propto \mathbb{E}_{f_*|\mathcal{D}, \pi} \left[ \mathbb{E}_{f_{\boldsymbol{x}}|\mathcal{D}, \pi}[\mathbb{E}_{y_{\boldsymbol{x}}|f_{\boldsymbol{x}}}[\log p(y_{\boldsymbol{x}}|f_{\boldsymbol{x}}, \pi, f_*)]]] - \mathbb{E}_{f_{\boldsymbol{x}}|\mathcal{D}, \pi}[\mathbb{E}_{y_{\boldsymbol{x}}|f_{\boldsymbol{x}}}[\log p(y_{\boldsymbol{x}}|f_{\boldsymbol{x}}, \pi)]] \tag{12}$$

$$\approx \frac{1}{Z_J} \sum_{j=1}^{J} \sum_{\ell=1}^{L} \sum_{k=1}^{K} \log p(y_{\boldsymbol{x}}^{(k)}|f_{\boldsymbol{x}}^{(\ell)}, f_*^{(j)}) \pi(f^{(\ell)}) \pi(f^{(j)}) - \sum_{\ell=1}^{L} \sum_{k=1}^{K} \log p(y_{\boldsymbol{x}}^{(k)}|f_{\boldsymbol{x}}^{(\ell)}) \pi(f^{(\ell)}), \tag{13}$$

where $Z_J$ is a normalizing constant $\sum_J \pi(f^{(j)})$ brought on by sampling optimal values, $y_{\boldsymbol{x}}|f_{\boldsymbol{x}}$ can trivially be obtained by sampling Gaussian noise $\varepsilon \sim \mathcal{N}(0, \sigma_{\varepsilon}^2)$ to a noiseless observation $f_{\boldsymbol{x}}|\mathcal{D}$ in the innermost expectation, and $f_{\boldsymbol{x}}$ and $f_*$ are obtained through the pathwise sampling procedure outlined in Eq. 6. The samples are evaluated on $p((y_{\boldsymbol{x}}|f_{\boldsymbol{x}}), (y_{\boldsymbol{x}}|f_{\boldsymbol{x}}, f_*))$. As evident by Eq. 13, $\pi$ notably affects the posterior distribution of both the observations $y_{\boldsymbol{x}}$ and the optimal values $f_*$. `PES` and `JES` are derived analogously. However, these acquisition function require conditioning on additional, simulated data and consequently, additional pathwise updates, to compute.

### 3.3 UPDATING THE USER BELIEF

Intuitively, a reasonable user should increasingly trust the data over their prior belief, as more data is acquired. For this purpose, we utilize a generalized likelihood (Grünwald, 2012; Friel & Pettitt, 2008) (a.k.a. power posterior or tempered likelihood) framework, commonly used in presence of model misspecification, to initially increase the impact of $\pi$ on the posterior predictive, and gradually decrease its impact over time. As such, we define the *tempered user belief* as

$$\pi(f; \beta, n) = \pi(f)^{\beta/n}, \tag{14}$$

where $\beta$ specifies the tempering rate and $n$ the iteration number. A similar approach of diminishing the impact of $\pi$ over time has been adopted in prior works (Hvarfner et al., 2022b; Souza et al., 2021). We note, however, that `ColaBO` differs in that the decay acts on the *model* in a principled Bayesian manner, as opposed to heuristically on the *acquisition function* in prior work.

### 3.4 PRACTICAL CONSIDERATIONS

`ColaBO` introduces additional flexibility to MC-based BO acquisition functions. The `ColaBO` framework deviates from vanilla (Q)MC acquisition functions (Wilson et al., 2017; Balandat et al., 2020) by utilizing approximate sample functions from the posterior, as opposed to pointwise draws from the posterior predictive and the reparametrization trick (Kingma & Welling, 2013; Rezende et al., 2014). `ColaBO` holds three shortcomings not prevalent in vanilla MC acquisition functions: (1) it cannot utilize Quasi-MC in the draws from the predictive posterior (only in the RFF weights), (2) it cannot fix the base samples (Balandat et al., 2020) drawn from the posterior for acquisition function consistency across the search space, and (3) the RFF approximation of $p(f)$ introduces bias. In Sec. 4.1, we empirically display the impact of these shortcomings. Notably, while acquisition function optimization no longer enjoys improved accuracy resulting from reparametrization, the acquisition function can still be effectively optimized due to the fact that `ColaBO` backpropagates through quantities computed as sums of smooth functions.

## 4 RESULTS

We evaluate the performance of `ColaBO` on various tasks, using priors over the optimum $\pi_{x_*}$ obtained from known optima on synthetic tasks, as well as from prior work (Mallik et al., 2023) on realistic tasks. We consider two variants of `ColaBO`: one using `LogEI` (Ament et al., 2023), a numerically stable, smoothed `logsumexp` transformation of `EI` with analogous derivation, and one variant using `MES`. We benchmark against the vanilla variants of each acquisition function, as well as $\pi$BO (Hvarfner et al., 2022b) and sampling from $\pi_{x_*}$. All acquisition functions are implemented in BoTorch (Balandat et al., 2020) using a Matérn-5/2 kernel (Matérn, 1960) and MAP hyperparameter estimation. Unless stated otherwise, `ColaBO` and $\pi$BO are initialized with the mode of the prior followed by 2 Sobol samples, whereas `LogEI` and `MES` are conventionally initialized with $D + 1$ Sobol samples. For all experiments, we run `ColaBO` with $\beta = N/2.5$, where $N$ is the total number of iterations, in order to make the prior influence approximately equal across experiments with differing budgets. We investigate the sensitivity to $\beta$ in Appendix B.3. The complete experimental setup is presented in detail in Appendix A, and our code is publicly available at https://anonymous.4open.science/r/colabo-6712/.

### 4.1 APPROXIMATION QUALITY OF THE COLABO FRAMEWORK

Firstly, we demonstrate the approximation quality of `ColaBO` *without* user priors to assert its accuracy compared to a vanilla MC acquisition function. To facilitate comparison, we randomly sample 10 points on the Hartmann (3D) function, and optimize `LogEI` with a large budget. We subsequently optimize `ColaBO`-`LogEI` on the same set of points and compare the $\arg\max$ to the solution found by the gold standard. Figure 3 displays the (log10) Euclidian distance between the $\arg\max$ of `LogEI` and its `ColaBO` variant. We note that, for small amounts ($\leqslant 256$) of posterior samples, the error induced by RFF bias is relatively low, which is evidenced by all RFF variants being roughly equal in distance to the true acquisition function optimizer.

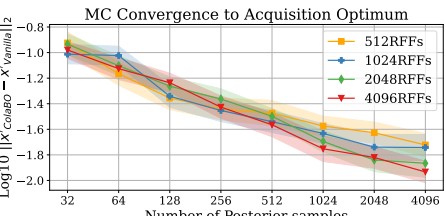

**Figure 3:** Mean and $1/4$ standard deviation of MC-induced errors of `ColaBO`-`LogEI` relative vanilla `LogEI` as measured by the distance to the $\arg\max$ of the acquisition function on Hartmann (3D) on 10 randomly sampled points for 40 seeds.

### 4.2 SYNTHETIC FUNCTIONS WITH KNOWN PRIORS

We adapt a similar evaluation protocol to Hvarfner et al. (2022b), and evaluate `ColaBO` for two types of user beliefs for synthetic tasks: well-located and poorly located priors over the optimal location, designed to emulate a well-informed and poorly-informed practitioner, respectively. The well-located prior is offset by a small (10%) amount from the optimum, and the poorly located prior is maximally offset, while retaining its mode inside the search space. Complete details on the priors can be found in Appendix A.3. On well-located priors, both `ColaBO`-`LogEI` and `ColaBO`-`MES` demonstrate substantially improved performance relative to their vanilla counterparts, comparable to $\pi$BO on all benchmarks. On poorly located priors, `ColaBO` demonstrates superior robustness, recovering the performance of the vanilla acquisition function within the maximal budget of $20D$ iterations and clearly outperforming $\pi$BO, which more frequently misled by the poor prior. In Appendix B.1, we also demonstrate `ColaBO` utilizing (accurate) beliefs over the optimal value: similarly to Figure 4, `ColaBO` yields increased efficiency relative to baselines, albeit not as substantial. Moreover, we demonstrate its usage with batch evaluations on well-located priors in Sec. B.2, showing that the drop in performance from batching evaluations is marginal at worst.

### 4.3 HYPERPARAMETER TUNING TASKS

Lastly, we evaluate `ColaBO` on three $4D$ deep learning HPO tasks from the PD1 (Wang et al., 2023) benchmarking suite. While the optima for these tasks are ultimately unknown, we utilize the priors

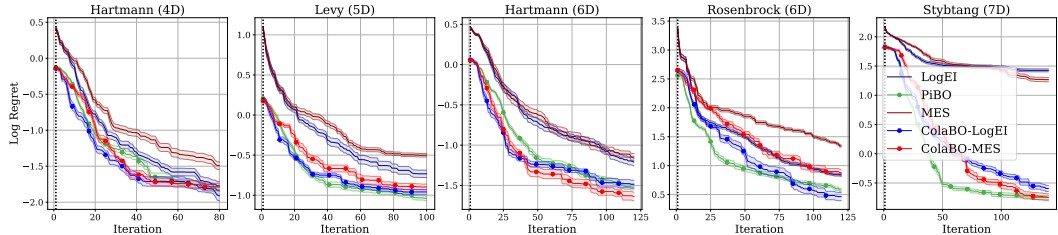

**Figure 4:** Performance on synthetic functions with well-located priors. Both `ColaBO-LogEI` and `ColaBO-MES` offer drastic speed-ups over their vanilla variants, and offer similar performance to $\pi$BO. The ranking of `ColaBO` acquisition functions are generally consistent with their respective vanilla variants. This is most prominent on Rosenbrock (6D), where `ColaBO-MES` struggles similarly to vanilla `MES`.

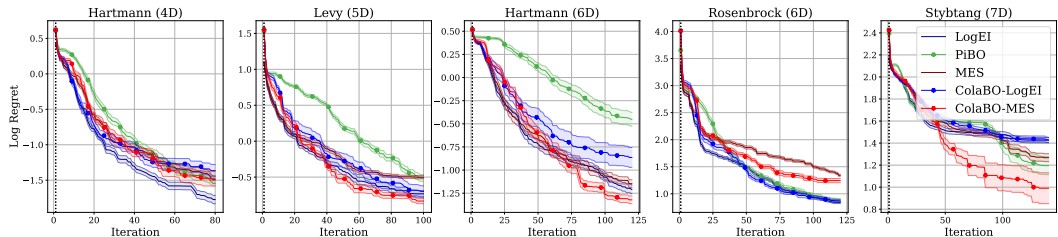

**Figure 5:** Performance on poorly located priors. `ColaBO` acquisition functions are more robust than $\pi$BO, as it frequently recovers the performance of the vanilla acquisition function before the total budget is depleted. `ColaBO-LogEI` struggles marginally on Hartmann (6D), whereas `ColaBO-MES` recovers or surpasses baseline performance on all tasks.

provided in MF-Prior-Bench [1] (Mallik et al., 2023), which are intended to provide a good starting point for further optimization. To emulate a realistic HPO setting, we consider a smaller optimization budget of $10D$ iterations, and initalize all methods that utilize user beliefs with only one DoE sample, that being the mode of the prior. The two `ColaBO` variants perform best in this evaluation, producing the best terminal performance on two tasks (WMT, LM1B), with all methods being tied on the third (CIFAR). `ColaBO` also finds highly competitive configurations in as few as ten evaluations. $\pi$BO offers similarly competitive performance, but does not offer the rapid initial progress of `ColaBO`, especially on LM1B. Crucially, `ColaBO` demonstrates substantial and

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

### B.3 $\beta$ SENSITIVITY ANALYSIS

We consider the effect of the $\beta$ hyperparameter of `ColaBO` introduced in Section 3.3, controlling the tempering of the prior. At first glance, a larger beta should force the optimizer to retain queries close to the prior mode for longer. However, having the optimum of function draws at the mode of the prior does not preclude other regions of the search space from being almost as good. As such, a very large $\beta$ can still allow for queries that are distant from the good region under the prior. However, a large beta does force very local queries early on, and require a larger amount of data to retract back to a conventional posterior. In Figs. 11, 9, 12 and 10, we display the performance of `ColaBO-LogEI` and `ColaBO-MES` for three values of beta and both well-located and misleading priors for a subset of the benchmarks from Sec. 4.2.

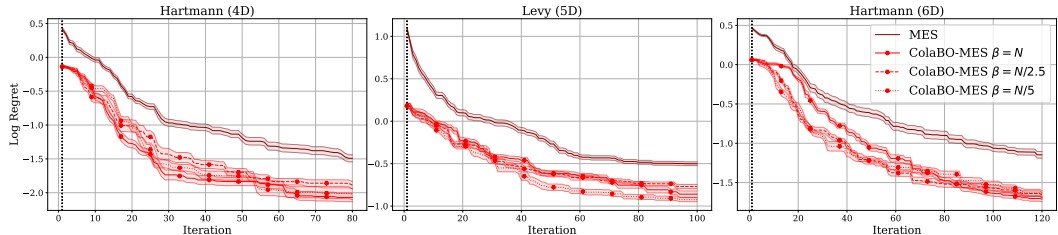

**Figure 9:** $\beta$ ablation on synthetic functions with well-located priors for `ColaBO-MES`. The magnitude of beta does not substantially impact performance, as rankings are not consistent across benchmarks. As the prior is not perfectly located, a smaller value of $\beta$ can reasonably yield a better terminal performance.

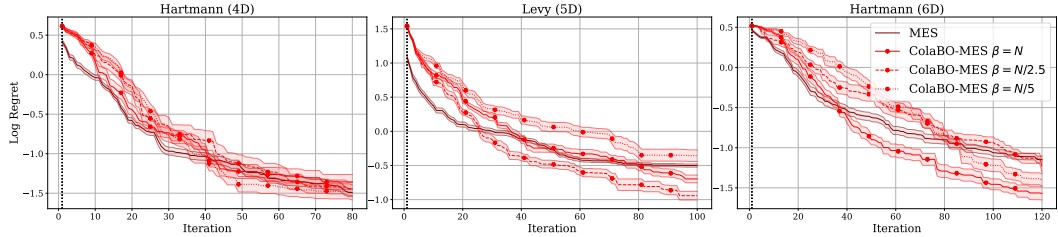

**Figure 10:** $\beta$ ablation on synthetic functions with poorly located priors for `ColaBO-MES`. `ColaBO-MES` displays very good robustness to poorly located priors, recovering approximately default performance for any $\beta$.

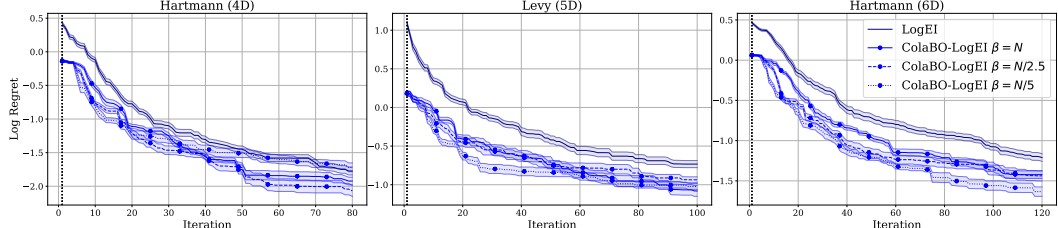

**Figure 11:** $\beta$ ablation on synthetic functions with well-located priors for `ColaBO-LogEI`. Similarly to `ColaBO-MES`, The magnitude of beta does not substantially impact performance, as rankings are not consistent across benchmarks. As the prior is not perfectly located, a smaller value of $\beta$ can yield a better terminal performance.

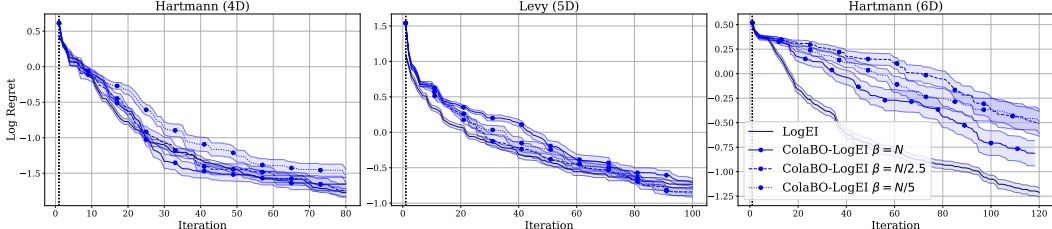

**Figure 12:** $\beta$ ablation on synthetic functions with poorly located priors for `ColaBO-LogEI`. The method displays good robustness to poorly located priors on two out of three benchmarks, recovering approximately default performance for any $\beta$. On Hartmann (6D), recovery is slower, and baseline performance is not recouped within the optimization budget for any of the tested $\beta$.