# OpenReview forum: "A General Framework for User-Guided Bayesian Optimization"
_ICLR.cc/2024/Conference — ICLR 2024 spotlight_

### Official Review · Reviewer_TtXD · 2023-10-29

**Soundness:** 3 good
**Presentation:** 3 good
**Contribution:** 2 fair
**Rating:** 6
**Confidence:** 3

**Summary:**

This submission provides a unifying framework for encompassing user beliefs in Bayesian Optimization (BO) beyond the usual priors on kernel  hyperparameters. Previous approaches offered to augment BO with expert beliefs $\pi$, like optimum value or location, but mostly focused on doing so at the acquisition function level. Here, the authors propose to integrate this at the surrogate level. Instead of the classical Gaussian Process prior $p(f)$, they introduce a user belief over functions $\pi(f) \propto \frac{p(f|\pi)}{p(f)}$. Thus, $p(f|\pi)$ can be obtained by reweighting samples from $p(f)$ proportionally to their probability of occurring under $\pi(f)$.

As the user belief is assumed independent of the data-generating mechanism, the resulting posterior $p(f|\mathcal{D},\pi)$ is naturally proportional to the user belief $\pi(f)$ and the likelihood $p(f|\mathcal{D})$. It is therefore non Gaussian for nontrivial user beliefs, an issue circumvented by the authors using a decoupled sampling scheme. Likewise, classical acquisition functions like Expected Improvement or Maximum Entropy Search are not tractable anymore, which led the authors to leverage and tailor their Monte-Carlo version to this specific case.

In the end, the proposed method, *ColaBO*, is then evaluated on a range of synthetic and real-world benchmarks, and demonstrates convincing performances against its competitors, particularly for misleading user beliefs.
As these benchmarks contain hyperparameter tuning of Deep Learning models, I believe this submission completely falls into the scope of ICLR.

**Strengths:**

- To the best of my knowledge, this is the first attempt to integrate user prior beliefs directly at the level of the surrogate rather than at the acquisition function level. The approach is novel and encompasses multiple ways for the user to incorporate its expertise: knowledge of function optimum value, optimum location, and preferences.
- I like Figures 1 and 2, they nicely illustrate the benefits of incorporating user beliefs and how this impacts the GP posterior and the acquisition function landscape.

**Weaknesses:**

I cannot think of any salient weakness in this work. *ColaBO* relies on several approximations due to non-Gaussianity of the posterior and these can probably be made more efficient, as mentioned in the limitations.

**Questions:**

I do not have questions.

Typos or similar:

- Can you clarify what "DoE" means in "[...] we consider a smaller optimization budget of $10D$ iterations, and initialize all methods that utilize user beliefs with only one DoE sample, that being the mode of the prior".
- Figure 6: The y-axis gives "Accuracy" but performances are decreasing over the BO trial. It probably should be something like 1-accuracy?
- A.3: "by using a sampling an offset direction $\boldsymbol{\epsilon}$ - > "by sampling an offset direction $\boldsymbol{\epsilon}$"?
- Concusion: "[...] or pre-trainedp" -> pre-trained

---

> ### Author Response · Authors · 2023-11-22
> **Response to reviewer TtXD**
>
> We thank the reviewer for their positive review. We have addressed the minor comments (except for the plots, which we will have to edit for the CR).
>
> Hopefully our new additions, mentioned in the global reply further enhances the reviewer's perception of our work. If so, we would greatly appreciate if the reviewer would increase their score, or confidence, to reflect this.

---

### Official Review · Reviewer_ysEZ · 2023-10-29

**Soundness:** 3 good
**Presentation:** 2 fair
**Contribution:** 3 good
**Rating:** 8
**Confidence:** 4

**Summary:**

This paper proposes a new approach to user (or prior) guided Bayesian optimization. Unlike previous approaches where the acquisition function was modified to incorporate priors, the paper proposes to sample from a modified posterior of the Gaussian process. This is achieved by combining rejection sampling with the recent line of work on path-wise conditioning.

**Strengths:**

* The proposed technique is original and interesting. It certainly appears more principled than previous approaches to incorporating expert knowledge.

**Weaknesses:**

* The mathematical derivation is too informal in some places, affecting clarity. I believe this needs to be improved. For instance, $\pi$ here represents a belief over functions. Then why is Eq. (4) a function that receives a point $x \in \mathcal{X}$? Shouldn't it be a probability distribution over functions? Also, Def 3.1 introduces a conditioning on $\pi$. I found this quite confusing. Is $\pi$ a density? a function? or a random variable?
* There are concerns about the fairness of the experiments. The authors state that "ColaBO and πBO are initialized with the mode of the prior followed by 2 Sobol samples, whereas LogEI and MES are conventionally initialized with D + 1 Sobol samples." Shouldn't logEI and MES also have been initialized on the mode of the prior? If we assume that such prior information is present in user-guided BO methods, it is fair to assume that conventional BO methods also have access to this information.
* The evaluations are okay but not extensive (especially compared to the piBO paper). I think this is an important point, given that the utility of user-guided BO methods can only be judged empirically on a case-study basis. This connects with my next concern.
* The paper lists a couple of different types of function priors in Section 3.1 but only seems to evaluate one type of such prior. It is, therefore, hard to judge how general this framework is for incorporating prior knowledge. Especially since the users state in the contributions that one can "incorporate arbitrary user knowledge."

### Major Comments
* The contribution statement is too general except for item 2. For instance, 1 and 3 can be applied to any user-guided BO method. I expect some more technical details about what this method offers to the field of user-guided BO.
* The derivation of importance sampling in Eq. (10) is unclear. $\pi(f) p(f | D)$ is an unnormalized density, but the integral is taken over it. Then, the equality with Eq. (9), doesn't hold since the expectation is not normalized.
* In Section 3.3, the authors introduce a tempering scheme based on the number of datapoints. Furthermore, they draw connections with generalized Bayesian inference (GBI). I think the authors should make it clear that the connection is very loose. A major characteristic of GBI is to find a statistically principled way to come up with the temperature, which is not the case here. It is particularly misleading as, in the last sentence of Section 3.3, the paper states that tempering is done so in a "principled Bayesian manner."

### Minor Comments
* Couldn't the name collaborative Bayesian optimization be misleading as to making people to think this method involves multiple users?
* Section 1 first sentence: Please cite classic papers that introduced BO for historical context.
* Figure 9-12: The boundary of the error bands makes it hard to distinguish between the plots. Please consider improving the visibility here.
* Section 2.1 equation: Generally, we aim to minimize the expectation of f or the regret. Saying that we minimize f, in the presence of noise, is too informal.
* Section 2.2 second from the last sentence: I think replacing "standard method" with "classic method" would be better here since we have an influx of more efficient methods.
* Eq (2): Has the "equivalent in distribution" sign been defined anywhere?
* Section 3.1 second paragraph second sentence: the sentence is incomplete.
* References: please consider making the reference more consistent and check the entries. Carl et al 2022b has a typo: joint entropy eearch -> joint entropy search; Jones et al. 2018 is missing the journal entry; Kingma and Welling 2013 was published in ICLR’14.

**Questions:**

* Given that the authors rely on GP posterior sampling, an obvious thing that could have been tried is Thompson sampling. Have the authors tried to include it?

---

> ### Author Response · Authors · 2023-11-22
> **Response to Reviewer ysEZ**
>
> We thank the reviewer for their thorough and informed feedback. We generally find the review to be insightful and have done our best to address the weaknesses in our work. In the newly uploaded pdf, we have:
> 1. Added five additional HPO tasks (6D) from LCBench. Hopefully, this tackles the reviewer's concerns regarding the experimental evaluation. Moreover, all methods now utilize the same $\text{Mode} + 2$ initialization, on all tasks. Lastly, we have added an illustrative example of how priors over preference relations can be used in ColaBO in a newly added Fig. 1 in the intro.
> 2. Worked on clarity in Sec. 3 throughout. We address specific feedback below, but this includes:
> 	- The role of $\pi$ and its conditioning - substituted for $\rho$ to separate
> 	- Densities and integrals - some missing proportionalities and normalizing constants
> 	- GBI relation removed.
> 3. Specified contributions to better distinguish ColaBO from other user-guided BO.
> 4. Minor comments/faults have been addressed.
> For a discussion regarding each point of feedback, see below:
>
> #### __The mathematical derivation is too informal in some places, affecting clarity. I believe this needs to be improved. For instance, here represents a belief over functions. Then why is Eq. (4) a function that receives a point ? Shouldn't it be a probability distribution over functions? Also, Def 3.1 introduces a conditioning on $\pi$. I found this quite confusing. Is $\pi$ a density? a function? or a random variable?__
>
> We see that the confusion is warranted, and have consequently re-hashed section 3.1. Hopefully, this improves the understanding and overall experience. $\pi (f)$ is replaced by $\rho (f)$. Moreover, the domains, co-domains and variable properties (i.e. randomness of $\rho$ is communicated throughout.
>
> The idea of Eq. 4 was to demonstrate how $\pi(x)$ can be expressed as a function not of the optimizer $x^*$, but as a function of $f$. We subsequently introduce notation where $\rho$ is a function of $f$ for some arbitrary function property, e.g. $x_*$ and $f_*$. We have now re-done the notation to clarify the inputs to each quantity and hope that this improves the overall clarity.
>
> #### __In Section 3.3, the authors introduce a tempering scheme based on the number of datapoints. Furthermore, they draw connections with generalized Bayesian inference (GBI). I think the authors should make it clear that the connection is very loose.__
>
> We agree with the reviewer, and that section is now entirely removed. In later experiments, we found that the tempering did not substantially improve performance, and since its "principled" nature was debatable, we removed it to obtain an overall cleaner approach. In practice, the data is sufficient to decrease the impact of a poor user belief.
>
> #### __The derivation of importance sampling in Eq. (10) is unclear. is an unnormalized density, but the integral is taken over it. Then, the equality with Eq. (9), doesn't hold since the expectation is not normalized.__
>
> Thanks for catching this typo. We missed a normalizing constant on $\pi$, which will not generally be tractable. We have added this normalizing constant and clarified accordingly. We have added a breif comment on the sampling, but will cover it in more detail in the CR when space permits.

---

> > ### Author Response · Authors · 2023-11-22
> > **Minors and questions**
> >
> > #### __Couldn't the name collaborative Bayesian optimization be misleading as to making people to think this method involves multiple users?__
> >
> > That is fair. We very much liked the name since the user can collaborate with BO by expressing any type of prior they have. But given your feedback, we will reconsider the name. We have not found a more appealing one as of now, though. Would you have a suggestion? If so, please let us know.
> >
> > #### __Section 1 first sentence: Please cite classic papers that introduced BO for historical context.__
> > Thanks, we added Mockus, Jones et al and Snoek et al in the very beginning of the intro.
> >
> > #### __Figure 9-12: The boundary of the error bands makes it hard to distinguish between the plots. Please consider improving the visibility here.__
> > We thank the reviewer for taking the time to go through the appendix. As the tempering is no longer part of the paper, these plots are removed. We do, however, agree with the reviewer's point.
> >
> > #### __Section 2.1 equation: Generally, we aim to minimize the expectation of f or the regret. Saying that we minimize f, in the presence of noise, is too informal.__
> >
> > We believe we are in line with convention using this problem formulation. Can the reviewer to point to an example in the literature paper that they believe is suitable for the noisy problem setting?
> >
> > #### __Section 2.2 second from the last sentence: I think replacing "standard method" with "classic method" would be better here since we have an influx of more efficient methods.__
> > We agree, it has been changed.
> >
> >
> >
> > #### __Eq (2): Has the "equivalent in distribution" sign been defined anywhere?__
> > That is correct, it has not. We have not added it as of now, and have not seen it defined e.g. in Wilson et. al (2020). Nonetheless, we will add it in the camera ready if the reviewer believes it is necessary (and spacing permits).
> >
> > #### __Section 3.1 second paragraph second sentence: the sentence is incomplete.__
> > We agree. Section 3 is rewritten, so that sentence is no longer present.
> >
> > #### __References: please consider making the reference more consistent and check the entries. Carl et al 2022b has a typo: joint entropy eearch -> joint entropy search; Jones et al. 2018 is missing the journal entry; Kingma and Welling 2013 was published in ICLR’14.__
> > Thanks for being thorough and catching these. We have updated accordingly.
> >
> > _____
> >
> > We thank the reviewer for the very thorough feedback, and hope that our added experiments and the re-haul of Sec. 3.1 improved the reviewer's perception of our paper. If there are any outstanding questions in the little time that remains of the rebuttal, we would be happy to address them.

---

> ### Comment · Reviewer_ysEZ · 2023-11-22
> **Response**
>
> I sincerely thank the authors for doing their best to address my concerns. I can imagine that the authors spent quite a tough few weeks!
>
>
>
> It is a little bit sad that making the experiments fairer took a toll on the significance of the proposed methodology. But still, it appears that the proposed framework has benefits, especially in terms of worst-case performance on the Hartmann 6 problem. Furthermore, I find the technique itself novel and interesting, as mentioned in my original review. In light of the revision, I am happy to raise my score.
>
> Some additional comments:
> * Still, most of the experiments are less than 10 dimensions. To increase the impact of the work, I recommend higher dimensional problems.
> * The reference formats are not very consistent. For example, some proceedings are in capitals, while some are in sentence cases. Some ArXiv papers have the ArXiv code, but some don't (Huang 2022, for example), and some have URLs while others don't. I recommend polishing the reference section.
> * Eq (6): Seems like using $x$ is a little confusing here. How about using $y$ to denote the function values?
>
> Lastly, it seems that the authors might have missed my following comment:
> > Given that the authors rely on GP posterior sampling, an obvious thing that could have been tried is Thompson sampling. Have the authors tried to include it?
> I would be interested to hear if the authors could run additional experiments with Thomson sampling. (Given that the rebuttal period is almost over, this is a friendly suggestion, not a request!)

---

> > ### Author Response · Authors · 2023-11-23
> > **Response and thanks**
> >
> > We thank the reviewer for the additional comments and their positivity towards our work! We will make sure to address the additional comments in the CR.
> >
> > The TS exclusion was an oversight, but it is now added (the decoupled variant, without user prior resampling, which it could be used in conjunction with). As there are many methods included who are relatively close (Fig. 6 in particular), we will do what we can to improve visibility in the CR.

---

> > > ### Comment · Reviewer_ysEZ · 2023-11-23
> > > **Thanks**
> > >
> > > > which it could be used in conjunction with).
> > >
> > > I was actually curious to see this one since it would be very easy to combine with the proposed method!

---

### Official Review · Reviewer_ZcGu · 2023-10-30

**Soundness:** 3 good
**Presentation:** 2 fair
**Contribution:** 3 good
**Rating:** 6
**Confidence:** 3

**Summary:**

The paper considers a setting where BO is applied to tackle black-box optimization problems where good prior knowledge is available. However, the conventional GP prior might fall short in effectively incorporating this knowledge. Therefore, the authors propose a new approach to inject it, mainly based on reweighing the prior of the GP with the user-defined prior and deterministic update of the GP posterior. Empirically, two instances of the framework is tested in synthetic and hyperparameter tuning tasks.

**Strengths:**

1.	The paper considers a novel black-box optimization setting where good prior knowledge exists and proposes a framework to handle this problem.
2.	Based on sampling, this framework is compatible with all Monte Carlo acquisition functions.
3.	With acquisitions to be Log Expected Improvement and Max-Value Entropy Search, the proposed models work well in synthetic and hyperparameter tuning tasks when well-located prior to the optimal location is available, while the drop in performance is not obvious compared to the benchmark models.

**Weaknesses:**

1.	Although the empirical performance of ColaBO looks promising in the synthetic task and hyperparameter tuning task, the theory developed in the work is limited. Therefore, how the method performs statistically is a concern, given that there are many approximations, such as the Monte Carlo acquisition and the RFF sampling.
2.	The empirical section can be further enhanced by testing the algorithms on more challenging tasks, for example, higher-dimensional problems. The paper primarily focuses on the low-dimensional tasks, which might not provide a strong basis for demonstrating the algorithm’s performance in realistic problems, where the dimensions might be much higher.
3.	The presentation of the work can be improved. Initially, the authors define $\pi(\cdot)$ in Eq. 3 with input from $\mathcal{X}$, but it later becomes $\pi(f)$. Moreover, the notations become messier after Eq. 4, making it more challenging to follow. Also, while rejection sampling seems to be an important step in the model, the way it works is not introduced in the paper. Similar for RFF.
4.	Given the authors’ claim that this is a general framework for BO to incorporate prior, the paper should present concrete examples to demonstrate its generality. For example, the authors could show how the examples in the second paragraph of the introduction section can be solved effectively within the proposed framework.

**Questions:**

1.	What is the difference between $f^*$ and $f_*$? What is $x_*$?
2.	Is there any reason the authors in favor of rejection sampling compared to other sampling methods? What is the efficiency of using rejection sampling here?
3.	How $\beta$ is determined?
4.	In Figure 5, even though ColaBO-MES is injected with a poorly located priors, it still performs better than MES, how do the authors interpret that?
5.	The authors show in Eq. 5 that $p(f|D, \pi) \propto \pi(f)p(f|D) $. Does this proportionality still hold in Eq. 6?
6.	To indicate where the optimum is located within the prior, couldn’t we achieve it by simply defining the prior mean function in GP properly? For instance, assigning higher values to points where the users believe to be good and low values to the rest.
7.	Why do the authors consider log EI instead of EI?

**Details Of Ethics Concerns:**

NA.

---

> ### Author Response · Authors · 2023-11-22
> **Response to Reviewer ZcGu**
>
> We thank the reviewer for their informed feedback. We agree with the reviewer on the outlined weaknesses, and have addressed them, incorporating the following in the newly uploaded manuscript:
>
>
> 1. We have added more HPO tasks (6D) from LCBench. Hopefully, this gives a more complete view of performance of ColaBO. They can now be found in the newly added Appendix A, but will go into Sec. 4 when space permits.
> 2. Introduced examples (similar to Fig. 1) of ColaBO with max-value priors and priors over preferences to better highlight how various tasks can be addressed within the ColaBO framework.
> 3. Re-worked notation in Sec. 3. The notation of $\pi(f)$ is gone, and replaced with $\rho(f)$ to better distinguish between them. Moreover, the properties of $\rho$ are clearly specified, and it is defined in what we believe is a more well-defined manner. However, we have re-hashed and re-worded the notation to hopefully improve the reading experience. While time is short, we would greatly appreciate additional feedback on how clarity can be improved here.
> 4. Previous Eqs. 8-13 have been clarified with regard to $f_*$, $x_*$ and the typo $f_n^*$.
> Below, we address each of the reviewer’s comments in detail.
>
> ## Responses to concerns:
> #### __Although the empirical performance of ColaBO looks promising in the synthetic task and hyperparameter tuning task, the theory developed in the work is limited.__
>
> We respect the reviewer's viewpoint and agree that the theoretical work is not extensive. However, the introduced theory serves the necessary purpose and yields a valid, conceptually simple and broadly applicable approach.
>
> Notably, there is a rather large collection of work in user-guided BO of various kinds (cited in the intro of the paper) which utilizes various types of knowledge, and ColaBO can successfully handle all of these types in a principled manner _without altering the approach_. As such, one can argue that this one piece of theory produces the general framework that user-guided BO has been missing. The newly added Fig. 1 exemplifies how all of these user beliefs are seamlessly dealt with by ColaBO.
>
> We acknowledge that work on scalability and approximation quality should be addressed moving forward, but ultimately view this line of work as follow-up.
>
> #### __The presentation of the work can be improved. Initially, the authors define in Eq. 3 with input from $\pi(x)$, but it later becomes $\pi(f)$. Moreover, the notations become messier after Eq. 4, making it more challenging to follow. Also, while rejection sampling seems to be an important step in the model, the way it works is not introduced in the paper. Similar for RFF.__
>
> We outline RFFs in the first paragraph of 2.3 and couldn’t find space to expand on it further. The exclusion of various sampling approaches (rejection, importance) but will be added to a camera-ready along with RFFs when space will permit it.
>
> We have made our best attempt at addressing your broader concern regarding clarity. Section 3.1 is now presented using slightly altered notation, where each property is properly distinguished and defined.
>
> #### __Given the authors’ claim that this is a general framework for BO to incorporate prior, the paper should present concrete examples to demonstrate its generality. For example, the authors could show how the examples in the second paragraph of the introduction section can be solved effectively within the proposed framework.__
>
> We agree that the conventional prior over the optimum is over-emphasized in the experiments we had presented. To the reviewer’s point, we have added Fig. 1 to highlight examples of how each type of prior is incorporated into ColaBO. Hopefully, this serves to highlight the generality of ColaBO, as requested by the reviewer.
>
> Moreover, there are max-value prior tasks in App. C.2, Fig. 10. We will try and highlight these types of tasks in the paper by adding yet another one.

---

> > ### Author Response · Authors · 2023-11-22
> > **Response to ZcGu, part 2**
> >
> > ### Answers to questions
> >
> > #### __What is the difference between__ $f^*$, $f_*$ __and__ $x^*$?
> > Thanks for pointing this out, we have defined these objects more clearly in the new pdf. $f_*$ is the optimal value, $x_*$ is the optimal location —  $f^*$ is a typo which we have now fixed. Thank you for catching this! We used subscripts (as opposed to superscripts) to avoid double superscripts.
> >
> > #### __Is there any reason the authors in favor of rejection sampling compared to other sampling methods?__
> > We initially pursued importance- and rejection sampling, but quickly found that rejection sampling yielded similar or better performance with substantially smaller memory usage, since many importance weights would become very small. Our method is, however, compatible with importance sampling.
> >
> > ##### What is the efficiency of using rejection sampling here?
> > This depends on the dimensionality and strength of the prior, but in our experiments on the order of one in a thousand at worst, for the higher-dimensional problems. The procedure of acquiring samples takes between 1 and 5 seconds depending on the problem.
> >
> > While this is certainly a drawback, the worst case only happens if the user specifies a narrow prior over each added dimension. We would reasonably expect the user to have a sound belief over a handful (<5) dimensions, and typically put a uniform prior on the rest. For this number of non-uniform dimensions _on the prior_, the sampling step does not bottleneck runtime.
> >
> > ##### __In Figure 5, even though ColaBO-MES is injected with a poorly located priors, it still performs better than MES, how do the authors interpret that?__
> > We posit that this is due to the RFF and MC approximations inducing additional randomness in the queries (caused by approximation errors), which assists in Stybtang specifically, where most acquisitions get stuck. As such, we do not believe ColaBO-MES is generally superior, and the outperformance is circumstantial.
> >
> > ##### __In Eq. 5, the authors show that p(f|D, \pi) \propto \pi(f)p(f|D) Does this proportionality still hold in Eq. 6?__
> > Eq. 6 follows from Eq. 5 and Matheron's rule, where the realization of a random variable (whether $p(\hat{f})$ or $p(\hat{f}|\pi)$) is updated deterministically using the update described in 6. Since $(\hat{f}|\pi) \sim p(\hat{f})\pi(\hat{f})$, this holds.
> >
> > ##### __To indicate where the optimum is located within the prior, couldn’t we achieve it by simply defining the prior mean function in GP properly? For instance, assigning higher values to points where the users believe to be good and low values to the rest.__
> >
> > Certainly - This has been done in Snoek et. al. (2015) for similar purposes. However, _it is unclear how a given mean function translates to a desired distribution over the optimum_, so one would have to properly parametrize and tune the mean function to get the desired distribution over the optimum. Moreover, the induced distribution over the optimum would be affected by both prior mean and lengthscales (and thus change when these estimates change) whereas ColaBO always retain the distribution over the optimum that has been specified, regardless of the hyperparameters of the model due to the orthogonality of these priors.
> >
> > Specifically, the approximated GP mean in Fig. 1 (bottom left, dark blue) is a result of the prior (green). The reviewer's approach does this process in the reverse order - the prior over the optimum would follow from the (parameterizable) mean, but the mean would need updating if other HPs change.
> >
> > Lastly, the approach would be orthogonal to ColaBO, in the sense that one could define _both_ a non-constant mean _and_ a prior over the optimum for the same problem. By aligning them, one would increase the efficiency of sampling (since more samples would have their optima at the mode of the user prior) but would introduce additional hyperparameters.
> >
> > ##### __Why do the authors consider log EI instead of EI?__
> > Apart from LogEI being superior from an acquisition optimization perspective (Ament et. al. 2023), smooth acquisition functions such as LogEI or MES are easier to approximate than non-smooth ones.
> >
> > We hope to have removed (some of) your concerns and would appreciate it if this gives you confidence to increase your score. While there is little time left of the rebuttal, we would be happy to discuss any follow-up questions or concerns.
> >
> > ### References:
> >
> > Sebastian Ament, Samuel Daulton, David Eriksson, Maximilian Balandat, and Eytan Bakshy. Unexpected improvements to expected improvement for bayesian optimization. In _Thirty-seventh Conference on Neural Information Processing Systems_, 2023.
> >
> > J. Snoek, O. Rippel, K. Swersky, R. Kiros, N. Satish, N. Sundaram, M. M. A. Patwary, Prabhat, and R. P.
> > Adams. Scalable Bayesian optimization using deep neural networks. In _Proceedings. of the International Conference on Machine Learning, 2015.

---

### Official Review · Reviewer_tGCg · 2023-10-31

**Soundness:** 3 good
**Presentation:** 3 good
**Contribution:** 3 good
**Rating:** 8
**Confidence:** 4

**Summary:**

The paper introduces ColaBO, which allows domain experts to customize the BO optimization process by integrating prior beliefs, such as information about the probable location of the optimum or the optimal value. Through empirical experiments, it shows that ColaBO speeds up optimization when prior information is accurate and maintains reasonable performance even when the prior knowledge is misleading.

**Strengths:**

The framework's adaptability and flexibility to incorporate prior knowledge into the optimization process. The method maintains reasonable performance even when the prior knowledge is misleading, demonstrating its robustness in different scenarios.

**Weaknesses:**

Test functions used to evaluate the proposed framework were quite limited, only a restricted set of test functions was employed.

**Questions:**

Besides the likely location of the optimizer or the optimal value, what other forms of prior knowledge would generally be useful?

---

> ### Author Response · Authors · 2023-11-22
> **Response to reviewer tGCg**
>
> We thank the reviewer for their positive review.
> #### __Test functions used to evaluate the proposed framework were quite limited, only a restricted set of test functions was employed.__
> We have added 5 6D HPO tasks from the LCBench HPO benchmarking suite.Hopefully, this alleviates the reviewer's concern and further demonstrates the empirical performance and flexibility of ColaBO.
>
> #### __Besides the likely location of the optimizer or the optimal value, what other forms of prior knowledge would generally be useful?__
> We have added an additional example using beliefs about preference relations in the new Fig. 1,, where the user specifies pairs of configurations and their belief over which one is likely to be superior of the two. In the figure, we show that this type of prior yields a uniquely shaped model, and further emphasizes the flexibility of the framework.
>
> _______
> We hope that the additional tasks and Figure 1, which highlights the versatility of ColaBO, has increased the reviewer's confidence in the quality of our paper. If the reviewer has any additional questions in the remaining time, we would be happy to address them.

---

### Author Response · Authors · 2023-11-22
**Response to reviewers**

Dear reviewers,

Thanks for being patient with our rebuttal. Several pieces of feedback were shared across reviewers, and we almost exclusively agreed with the reviewers on their feedback. As such, the following changes have been made to the paper:

1. Added five additional HPO tasks (6D) from LCBench (App. A, will go in main paper in CR). Hopefully, this tackles the reviewers' shared concerns regarding the experimental evaluation. Moreover, all methods now utilize the same $\text{Mode} + 2$ initialization, on all tasks.
2. We have added an illustrative example of how priors over preference relations can be used in ColaBO in a newly added Fig. 1 in the intro to showcase how its versatility can generally be used.
2. Worked on clarity in Sec. 3 throughout. We address specific feedback below, but this includes:
	- The role of $\pi$ and its conditioning - $\pi(f)$ substituted for $\rho(f)$ to better distinguish
	- Densities and integrals -  missing proportionalities and normalizing constants
	- GBI and tempering  removed altogether - it cluttered performance and was not substantially benefitting performance.
3. Specified contributions to better distinguish ColaBO from other user-guided BO.
4. Minor comments/faults have been addressed.
5. For stability of the RFF approximation, RBF kernel is used instead of Matern in Sec. 4 and clearly stated in Sec. 3.3 (practical considerations), but Matern results are still presented in App. C2. Results are comparable between the two kernels.

---

### Meta-Review · Area_Chair_RxMZ · 2023-12-07

**Metareview:**

The authors consider the use of Bayesian optimization in a collaborative environment comprising a human-computer team. The authors propose "ColaBO," a framework to enable the user (assumed to be a domain expert but not necessarily an expert in machine learning/BayesOpt) to express beliefs about the objective function, such as likely locations for/values of the global optimum, in an approachable manner. In a series of experiments, the authors demonstrate that ColaBO is able to significantly accelerate optimization when this prior information is accessible.

Although there was some initial disagreement, the reviewers came to consensus in their recommendation of acceptance following the author response and reviewer discussion periods. In their reviews, the reviewers noted the importance of the problem setting and enthusiasm for the approach. I believe this will be a valuable addition to the ICLR program.

The reviewers did note some confusion regarding some minor points in the manuscript, many/most of which were sufficiently addressed during the author response period. I encourage the authors to take these initial points of confusion into account when revising their manuscript.

**Justification For Why Not Higher Score:**

Although universal, support from the reviewers was perhaps not quite strong enough to warrant an oral presentation.

Additionally, ignoring questions of "prestige" in oral vs spotlight presentation, I believe that this particular work is probably best incorporated in the program with a spotlight advertising the main idea (I believe people can become quickly excited here) followed by open-ended discussion at a poster session.

**Justification For Why Not Lower Score:**

In addition to universal support from the reviewers, the general area of human-computer teaming is relatively underrepresented at ICLR, and this work would increase the topical diversity of the spotlight presentations.

---

### Decision · Program_Chairs · 2024-01-16

Accept (spotlight)